



# Climate and ice sheet evolutions from the last glacial maximum to the pre-industrial period with an ice sheet – climate coupled model

Aurélien Quiquet[1,2], Didier M. Roche[1,2], Christophe Dumas[1], Nathaëlle Bouttes[1], and Fanny Lhardy[1]

[1]Laboratoire des Sciences du Climat et de l'Environnement, LSCE/IPSL, CEA-CNRS-UVSQ, Université Paris-Saclay, F-91191 Gif-sur-Yvette, France
[2]Earth and Climate Cluster, Faculty of Earth and Life Sciences, Vrije Universiteit Amsterdam, Amsterdam, the Netherlands

**Correspondence:** A. Quiquet (aurelien.quiquet@lsce.ipsl.fr)

**Abstract.**

The last deglaciation offers an unique opportunity to understand the climate – ice sheet interactions in a global warming context. In this paper, to tackle this question, we use an Earth system model of intermediate complexity coupled to an ice sheet model covering the Northern Hemisphere to simulate the last deglaciation and the Holocene (26-0 ka BP). We use a synchronous coupling every year between the ice sheet and the rest of the climate system and we ensure a closed water cycle considering the release of freshwater flux to the ocean due to ice sheet melting. Our reference experiment displays a gradual warming in response to the forcings, with no abrupt changes. In this case, while the amplitude of the freshwater flux to the ocean induced by ice sheet retreat is realistic, it is sufficient to shut down the Atlantic meridional overturning from which the model does not recover within the time period simulated. However, with reduced freshwater flux we are nonetheless able to obtain different oceanic circulation evolutions, including some abrupt transitions between shut-down and active circulation states in the course of the deglaciation. The fast oceanic circulation recoveries lead to abrupt warming phases in Greenland. Our simulated ice sheet geometry evolution is in overall good agreement with available global reconstructions, even though the abrupt sea level rise at 14.6 kaBP is underestimated, possibly because the climate model underestimates the millenial-scale temperature variability. In the course of the deglaciation, large-scale grounding line instabilities are simulated both for the Eurasian and North American ice sheets. The first instability occurs in the Barents-Kara seas for the Eurasian ice sheet at 14.5 kaBP. A second grounding line instability occurs circa 12 kaBP in the proglacial lake that formed at the southern margin of the North American ice sheet. With additional asynchronously coupled experiments, we assess the sensitivity of our results to different ice sheet model choices related to surface and sub-shelf mass balance, ice deformation and grounding line representation. While the ice sheet evolutions differ within this ensemble, the global climate trajectory is only weakly affected by these choices. In our experiments, only the abrupt shifts in the oceanic circulation due to freshwater fluxes are able to produce some millenial-scale variability since no self-generating abrupt transitions are simulated without these fluxes.

## 1 Introduction

The Quaternary has been marked by large sea level oscillations. A gradual sea level fall, associated with an increase in the continental ice sheet volume, characterises prolonged glacial periods lasting for several tens of thousand of years. In turn, short



glacial terminations precede interglacial periods that show reduced ice sheets. The study of glacial terminations can help us to understand the mechanisms behind large scale ice sheet retreat but also the key role of ice sheets within the global climate system.

During the last deglaciation (~21-7 kaBP), the sea level rose by about 120 metres to reach approximately its present-day level
(Waelbroeck et al., 2002; Lambeck et al., 2014). This rise is mostly explained by the disintegration of the North American and Eurasian ice sheets while Greenland and Antarctica together probably contributed to less than 20 metres (Whitehouse et al., 2012; Briggs et al., 2014; Lecavalier et al., 2014). The extent of the Northern Hemisphere ice sheets across the deglaciation is relatively well known although it can sometimes present large (>1 kyrs) dating uncertainties (Hughes et al., 2016; Dalton et al., 2020). However, the volume evolution of the individual ice sheets remains weakly constrained. In particular, sea level
archives have suggested the presence of abrupt sea level rises standing out from the gradual sea level rise of the deglaciation (Deschamps et al., 2012; Abdul et al., 2016; Harrison et al., 2019). These so-called melt water pulses suggest large-scale ice sheet instabilities but the contribution of the different ice sheets to these events remains debated (e.g. Liu et al., 2016).

Parallel to the non-linear ice sheet retreat, the atmosphere and the ocean have also undergone some large and abrupt variations.
For example, while atmospheric temperatures above Greenland rise gradually since the last glacial maximum (LGM), at the onset of the Bølling-Allerød period at 14.7 kaBP, they rise abruptly by more than 10 °C in a few decades (Severinghaus and Brook, 1999; Buizert et al., 2014). After 500 years of interglacial conditions, the climate abruptly returns to a cold state during the Younger Dryas (Alley, 2000a) from which the temperatures rise again steadily to reach their Holocene values. The evolution of the oceanic conditions are more uncertain. It seems nonetheless that the North Atlantic Deep Water (NADW) was
shallower at the LGM compared to today (Curry and Oppo, 2005). The 3D evolution of the water masses across the deglaciation is difficult to constrain given that different proxy can provide conflicting information (Waelbroeck et al., 2019). However, it is likely that the Atlantic meridional oceanic circulation (AMOC) has not remained constant, with possible rapid transitions from different states: intense, reduced, or even shut down (e.g. McManus et al., 2004; Ng et al., 2018).

The succession of events linking the changes in the atmosphere, ocean and ice sheets have yet to be formalised. Bi-directionally coupled ice sheet – climate models are ideal tools to study these interactions since they can explicitly represent the different climatic feedbacks at play, without having to prescribe ad-hoc external scenarios. In such coupling, the climate model provides the climatic forcing fields needed by the ice sheet model and in turns the ice sheet model provides an updated surface topography and ice sheet mask. Several coupled ice sheet – climate models are now available in the literature, spanning a
range of complexities. Given that the ice sheet integrates climate change over long timescales (> 10 kyrs), the vast majority of the work that has investigated multi-millenial climate change during the Quaternary has used simplified climate models to reduce the numerical cost (e.g. Calov et al., 2005; Fyke et al., 2011; Huybrechts et al., 2011; Heinemann et al., 2014). However, some general circulation model (GCMs) have also been bi-directionally coupled to ice sheet models (e.g. Vizcaíno et al., 2008; Gregory et al., 2012). In this case, the model is run for short integrations (typically less than a thousand year) or use



an asynchronous coupling to speed up the simulations (e.g. Ziemen et al., 2019). With the asynchronous coupling, the climate model is run less frequently than the ice sheet model (e.g. one year of climate is used to perform 10 years of ice sheet evolution).

To date, although a fair amount of coupled ice sheet – climate models exist, only few have been used to simulate the last deglaciation of Northern Hemisphere ice sheets. Thanks to an inexpensive set-up in term of computational cost, the CLIMBER-
2 Earth system model of intermediate complexity coupled to the SICOPOLIS ice sheet model has been used in several studies to simulate the last glacial-interglacial cycles (e.g. Ganopolski and Brovkin, 2017) and beyond (Willeit et al., 2019). CLIMBER-2 has also been coupled to alternative ice sheet models (Charbit et al., 2005; Bonelli et al., 2009). These studies have demonstrated the ability of the model to reproduce the global eustatic sea level reconstructions. They have also brought major improvements in our understanding of the respective role of orbital forcing, greenhouse gas mixing ratio, ice sheets and dust to explain the
past climatic variability. However, CLIMBER-2 shows drastic simplifications of the physics of the atmosphere (statistical-dynamical model on a coarse grid of 10°x~51° resolution) and in the ocean (three zonally averaged oceanic basin). Heinemann et al. (2014) used an alternative Earth system model of intermediate complexity, LOVECLIM (Goosse et al., 2010), to simulate the last deglaciation of Northern Hemisphere ice sheets. Compared to CLIMBER-2, LOVECLIM shows a higher spatial resolution in the atmosphere (~5.6°x5.6° resolution) and accounts for a general circulation oceanic model (Goosse and Fichefet,
1999). To successfully reproduce the ice sheet evolution Heinemann et al. (2014) have to use a correction of the climatic fields (namely temperature and precipitation). In addition, they use an asynchronous coupling to speed up their simulations. In doing so, they discard the role of freshwater flux to the ocean resulting from ice sheet melting. To our knowledge, no other bi-directionally coupled ice sheet – climate model has been used to simulate the last deglaciation of Northern Hemisphere ice sheets.


Building up on the work of Roche et al. (2014a), we present here the first comprehensive climatic simulations of the last deglaciation with interactive Northern Hemisphere ice sheets using a bi-directional synchronous coupling. We have performed different experiments with varying oceanic conditions to assess their importance in shaping the deglaciation. In addition, we have performed additional sensitivity experiments using an asynchronous coupling to assess the importance of some modelling
choices on our results. In Sect. 2 we present our model, the coupling strategy and the experimental setup. We show our results in terms of atmospheric temperature evolution, oceanic circulation changes and simulated ice sheets in Sect. 3. We discuss further our model limitations and foreseen improvements in Sect. 4 and conclude in Sect. 5.

## 2 Methods

### 2.1 Climate and ice sheet models

*i*LOVECLIM (here in version 1.1) is a code fork of the LOVECLIM 1.2 model (Goosse et al., 2010). The core of the model is a combination of a quasi-geostrophic atmospheric model solved on a T21 (~5.6°x5.6°) spectral grid (ECBilt, Haarsma





et al., 1997; Opsteegh et al., 1998); a free surface oceanic general circulation model on a 3°x3°spherical grid which includes a thermodynamic sea ice model (CLIO, Goosse and Fichefet, 1999); and a dynamic vegetation and carbon allocation model

(VECODE, Brovkin et al., 1997). *i*LOVECLIM has been extensively used to study millenial climate change during the Quaternary. For example, it has proven able to reproduce the glacial-interglacial variability of the hydrological cycle in the Tropics (Caley et al., 2014). It has also been used to study Heinrich events during the last glacial period (Roche et al., 2014b) or to investigate the processes responsible for changes in the carbon cycle during the last eight interglacial periods (Bouttes et al., 2018). With a similar model configuration to the one used in this work, *i*LOVECLIM results were included in the fourth phase

of the Palaeoclimate Modelling Intercomparison Project (PMIP) contribution to the Coupled Model Intercomparison Project (CMIP) (Kageyama et al., 2020).

Since Roche et al. (2014a), the model also includes a 3D thermomechanically coupled ice sheet model (GRISLI Ritz et al., 2001; Quiquet et al., 2018a). GRISLI solves the ice sheet mass conservation equation on a Cartesian grid. Like most ice

sheet model, deformation is computed with a Glen flow law in which anisotropy is artificially accounted for using an flow enhancement factor ($E_f$) that facilitates deformation induced by vertical shear. For the entire domain, the velocity field is the sum of velocity driven by vertical shearing (Shallow Ice Approximation, SIA) and the velocity driven by horizontal shearing (Shallow Shelf Approximation, SSA). In doing so, the SSA is used as a sliding law (Bueler and Brown, 2009; Winkelmann et al., 2011). Basal dragging $\tau_b$ is assumed to follow a linear friction law:

$$\tau_b = -\beta \mathbf{u}_b \tag{1}$$

where $\beta$ is the basal drag coefficient and $u_b$ is the velocity at the base. Cold based grid points have a virtually infinite friction at the base ($5 \times 10^5$ Pa yr m$^{-1}$), while floating ice shelves have no friction. For grid points at the pressure melting point, we use a friction computed from the effective pressure at the base of the ice sheet $N$:

$$\beta = c_f N \tag{2}$$

with $c_f$ a parameter that has to be calibrated. For the experiments shown here, we impose an ice flux at the grounding line that follows the analytical solution of Tsai et al. (2015). Calving at the ice shelf edge occurs if the ice thickness falls below a critical threshold and if the upstream Lagrangian ice flux does not allow to maintain an ice thickness above this threshold. The threshold is set here to 250 metres. Ice sheet model parameters (enhancement factor, basal drag coefficient and hydraulic conductivity) are calibrated in the same way as in Quiquet et al. (2018a) to reproduce glacial-interglacial Antarctic ice sheet

grounding line migration. In addition, we used a map of sediment thickness (Laske and Masters, 1997) to locally reduce basal dragging. We assume that for a sediment thickness greater than 200 metres, the basal drag coefficient in Eq. 2 is multiplied by a dimensionless factor of 0.05. The ice sheet model is run here on a Cartesian 40-km grid of the Northern Hemisphere using a Lambert azimuthal equal-area projection.





## 2.2 Ice sheet model coupling

The inclusion of GRISLI into iLOVECLIM has been presented in Roche et al. (2014a). However, the coupling procedure has been largely modified from this work. In particular, we have substantially improved the computation of surface and sub-shelf mass balance. Water conservation between GRISLI and the rest of the climate model has also been considerably improved. Details on this coupling is given in the following while its schematic representation is shown in Fig. 1. It is important to mention that only the Northern Hemisphere ice sheets are interactively simulated, while the Antarctic ice sheet topography and ice
mask remains prescribed.

### 2.2.1   Surface mass balance

In Roche et al. (2014a), the ice sheet surface mass balance (SMB) was computed from the annual mean precipitation and the annual and July mean near-surface air temperature using a positive degree day method (Reeh, 1989). Although computationally
inexpensive and easy to implement in a model, this method does not account for some important physical quantities that influence the SMB. In particular, the surface shortwave radiation is only implicitly taken into account, through the temperature. Instead, we use here the insolation temperature melt method (ITM) following Pollard (1980) and van den Berg et al. (2008). The amount of melt $M_s$ over one time step $dt$ is in this case:

$$M_s = max \left( \frac{dt}{\rho_w L_m} \left( (1-\alpha) SW_s + c_{rad} + \lambda T_s \right), 0 \right) \tag{3}$$

With $T_s$ is the near surface air temperature, $SW_s$ is the shortwave radiation at the surface, $\alpha$ is the surface albedo, $\rho_w$ is the density of liquid water and $L_m$ is the specific latent heat of fusion. $\lambda$ and $c_{rad}$ are empirical parameters that need calibration. In the literature, this calibration has been performed on observations of present-day glaciers. The $\lambda$ parameter is generally set to 10 W m$^{-2}$K$^{-1}$ (Pollard, 1980; van den Berg et al., 2008; Robinson et al., 2010). The parameter $c_{rad}$ is less constrained and is adjusted for the region considered (van den Berg et al., 2008). It is set to -50 W m$^{-2}$ in Pollard (1980), it ranges from -40 to
-60 W m$^{-2}$ in Robinson et al. (2010), whilst it is equal to -117 W m$^{-2}$ in van den Berg et al. (2008). We used $\lambda =$10 W m$^{-2}$K$^{-1}$ and $c_{rad} =$-40 W m$^{-2}$. However, iLOVECLIM presents an important warm bias in Eastern North America and a cold bias in Northern Europe that lead to unrealistic simulated ice sheet under glacial forcing. A problem also identified in Heinemann et al. (2014). To account for this, we use a local modification of the melt parameter $c_{rad}$ to partially correct these temperature biases. To this aim, we compute the annual mean temperature bias with respect to ERA-interim (Dee et al., 2011) and use a
linear correction in which a +10°C bias leads to $c_{rad} =$-80 W m$^{-2}$ (instead of the reference value of -40 W m$^{-2}$).

Because of the gap between the coarse atmospheric model resolution and the ice sheet model resolution, the downscaling of the forcing fields needed by the ice sheet model is a persistent issue in ice sheet – climate coupling. Here, we make use of the online dynamical downscaling embedded in iLOVECLIM (Quiquet et al., 2018b). This allows for the computation on
every atmospheric model timestep (4 hours) of snow, rain and near-surface air temperature at the ice sheet model resolution, explicitly taking into account the high-resolution topography. We used directly these fields to compute a surface mass balance





at the resolution of the ice sheet model with the ITM method (Eq. 3). The near-surface air temperature and the SMB are accumulated along the course of the year to generate the yearly forcing fields required by the ice sheet model. We made a few adjustments compared to the downscaling procedure presented in Quiquet et al. (2018b). In particular, some large-scale climate fields are now bi-linearly interpolated onto the high-resolution grid before the energy and moisture computation. This prevents the strong discontinuities that could exist between two sub-grid points belonging to two different large-scale grid cells.

### 2.2.2 Sub-shelf melt rate

The sub-shelf melt rate in Roche et al. (2014a) was imposed arbitrarily to an homogeneous and constant value for the entire Northern Hemisphere. Instead, we use here a physically based computation of the sub-shelf melt rate following Beckmann and Goosse (2003). For each vertical oceanic layer, $z$, we estimate the potential sub-shelf melt rate as:

$$M_{shelf}(z) = \frac{\rho_w c_p \gamma_T F_g TF(z)}{\rho_i L_m} \tag{4}$$

wit $c_p$ specific heat capacity of sea water, $\rho_i$ is the density of ice, $\gamma_T$ is the thermal exchange velocity and $TF(z)$ is the thermal forcing at depth $z$, defined as the difference between the ambient temperature and the temperature of the salinity dependent freezing point. $F_g$ is a weakly constrained dimensionless parameter and can be changed to explore the response of the ice sheet to different sub-shelf melt sensitivities to oceanic temperature change. In our reference experiment, we chose a parameter value ($15 \times 10^{-3}$) that produce about 0.1 m yr$^{-1}$ in the Arctic, a value similar to what is experiencing the Ross ice shelf today in Antarctica. In addition, in order to avoid unrealistic ice shelf expansion over the deep ocean we also impose a high sub-shelf melt rate of 20 m yr$^{-1}$ where the bathymetry is greater than 1500 m. Also, to mimic the fact that observed melt rates are greater in the vicinity of the grounding line, we double the value of the inferred melt rate in Eq. 4 for the floating points that are in contact with the grounding line. Eq. 4 is computed for each oceanic timestep (one day) and integrated over the year in order to provide the yearly forcing needed by the ice sheet model. There is no downscaling of the sub-shelft melt rate to the high-resolution ice sheet model grid, except that the depth of the ice shelf draft is used to determine the vertical layer $z$ in Eq. 4 that produces the melt.

### 2.2.3 Ice sheet feedbacks

Changes in the ice sheet feed back to the atmospheric and to the oceanic models. On the one hand, at the beginning of each year in the climate model, the ice mask and the orography in the climate model are changed according to the changes computed by the ice sheet model in the previous year. Both fields are aggregated from the ice sheet model resolution (40 km) to the T21 resolution in the same way as in Roche et al. (2014a). There is no partially glaciated grid cell in the atmospheric model: a coarse grid cell is considered as glaciated (ice mask set to 1) if it contains at least 30% of sub-grid points with an ice thickness greater than one metre. The ice mask in the atmospheric model impacts the surface albedo.





On the other hand, freshwater fluxes resulting from the ice sheet melting are transferred to the oceanic model. In Roche et al.
(2014a), the total ice sheet volume variation was transferred to the continental routing scheme assuming a uniform distribution
over the ice sheet. Only the calving flux was separated from the total volume variation to eventually feed an iceberg model
(Bügelmayer et al., 2015). This method has the advantage to ensure a closed water budget within the model but the spatial
information about ice sheet runoff is lost. For this reason, we now explicitly separate the different components of the global
volume variation on the ice sheet model side. Basal and surface melt of the grounded part of the ice sheet are transferred to the
routing scheme exactly where they occur. The basal melt below the ice shelves are also added to the ocean where they occur
but at the surface and not at depth. The calving flux can be either considered as the the basal melt, or used to feed the iceberg
model. At present, the iceberg model is not activated in our experiments and the calving flux, similarly to the sub-shelf melt, is
given at the oceanic surface. Local latent heat release resulting from iceberg melting is taken into account. Since the ice sheet
model main time step is one year, we do not have access to the seasonal cycle of the freshwater fluxes and their annual value
computed, by the ice sheet model, is homogeneously distributed through the year in the oceanic model.

## 2.3 Experimental setup

### 2.3.1 Boundary and initial conditions

The climate model uses time-varying information of greenhouse gases (Lüthi et al., 2008) and insolation (Berger, 1978). The
carbon dioxide mixing ratio evolution and the 65°N insolation in June is depicted in Fig. 3. We use a recent implementation
of the last glacial maximum bathymetry at 21 kaBP (Lhardy et al., 2020), which is left unchanged for the duration of the
experiments. Topography and ice mask are both provided by the ice sheet model.

To define our initial state, we run uncoupled ice sheet and climate experiments. First, we run the climate model using the last
glacial maximum boundary conditions for 3000 years. In this case, the ice sheet topography and ice mask correspond to the one
of the GLAC-1D reconstruction (Tarasov et al., 2012; Tarasov and Peltier, 2002; Briggs et al., 2014) at 21 kaBP. The different
experiments presented in the rest of the manuscript are all branched from the simulated climate at the end of this 3000 years. In
addition, the last 100 years of this experiment are also used to define a climatological annual surface mass balance and surface
temperature. We use this climatology to perform stand-alone ice sheet experiments starting from an ice free configuration of
the Northern Hemisphere. The ice sheet model is run for 200 kyr so that it has time to build-up the last glacial maximum ice
sheets. The simulated ice sheets after this spin-up are presented in Fig. 2a. The extent of the ice sheets agrees generally well
with the geologically constrained reconstruction of GLAC-1D (Fig. 2c) and ICE-6G_C (Fig. 2d, Argus et al., 2014; Peltier
et al., 2015) even though it is underestimated in the western part of the Eurasian ice sheet. The climate fields used to build
the spun-up ice sheets have been elaborated from the climate model with prescribed GLAC-1D boundary condition. As such,
the spun-up ice sheets should resemble the GLAC-1D reconstructions. If this is generally the case, there is nonetheless an
overestimation of the surface elevation of the North American ice sheet. This could indicate a precipitation overestimation in



this area or an underestimation of the ice sheet velocities. However, the fact that the Eurasian ice sheet does not present this bias points towards an overestimation of the precipitation. The spun-up ice sheets are used as initial conditions for the ice sheet model in the coupled experiments presented in this manuscript.

225

### 2.3.2 Description of the experiments

With have performed two sets of experiments to investigate two important points for the simulation of the deglaciation. First, the freshwater flux resulting from ice sheet melting likely influenced the climate evolution during the deglaciation since they can have lead to abrupt AMOC changes (e.g. Liu et al., 2009; Menviel et al., 2011; Obase and Abe-Ouchi, 2019). Thus, in a first set of experiments, we have performed various synchronously coupled experiments with varying oceanic circulation evolution. Second, several modelling choices related to the ice sheet model are not well constrained and could also have an influence on the simulated deglaciation. To tackle this problem, in a second set of experiments, we have performed various sensitivity experiments using an asyncronous coupling to reduce the computation cost. More details on these experiments are given in the following.

235

Our reference experiment (DGL) is an ice sheet – climate experiment, synchronously coupled. This experiment starts at 26 kaBP and uses the initial conditions presented in Sect. 2.3.1. The climate and the ice sheets used as initial conditions are not fully consistent between each other since they have been obtained with uncoupled long-term equilibriums. As such, the first thousand years or so of our experiments have to be discussed with care since part of the response can arise from artefacts due to the start of the coupling.

In addition to this reference experiment, we have performed additional synchronously coupled experiments for the first set of experiments which aims at investigating the importance of oceanic changes in shaping the last deglaciation.

First, we have run experiments in which the amount of freshwater is reduced in order to gradually limit their influence. In DGL_FWF/2 and DGL_FWF/3 we divide the flux resulting from ice sheet melting by two, respectively three, while in DGL_noFWF this flux is not injected to the ocean.

Second, it has been shown that the simulated NADW at the LGM in the iLOVECLIM model is too deep with respect to what suggest oceanic tracers (Lhardy et al., 2020), a feature shared with other PMIP participating models (Kageyama et al., 2020). This bias in the oceanic circulation can affect our results for the deglaciation. One way to provide an alternative oceanic circulation in the model is to use a parametrisation for the sinking of brines (Bouttes et al., 2010) around Antarctica. In this parametrisation, a fraction of the salt rejected by sea-ice formation (40%) is transferred to the deepest oceanic layer. This is done to artificially reproduce the sinking of dense waters induced by sea-ice formation along the continental slope of Antarctica since such process cannot be properly resolved in a 3°x3°resolution oceanic model. The parametrisation favours vertical stratification around Antarctica, enhancing Antarctic Bottom Water (AABW) and conversely weakening and shallowing of the NADW. Under glacial conditions, this leads to a better agreement with palaeo-data (Lhardy et al., 2020). We have thus





performed an experiment in which the parametrisation for the sinking of brines is activated (DGL_brines). The experiments with reduced freshwater flux and with the parametrisation for the sinking of brines are branched from the reference experiment DGL at 21 kaBP. At that time the ice sheets are not contributing to sea level change (total mass change of zero).

260 The second set of experiments consist of asynchronously coupled experiments to assess the sensitivity of our results to the modelling choices for the ice sheet model. In these experiments, the forcings (greenhouse gas mixing ratio and orbital forcing) are accelerated with a factor of 5. Acceleration has already been used extensively in the literature (e.g. Jackson and Broccoli, 2003; Gregory et al., 2012; Roberts et al., 2014; Heinemann et al., 2014; Choudhury et al., 2020). The accelerated experiments cover the 26-0 kaBP timespan but only 5200 years are computed in the climate model instead of the full 26000 years. In such 265 experiments, the ice sheet model is run for 5 years after 1 year of simulated climate so that only the ice sheet forcings are accelerated but not ice dynamics. This method allows to significantly reduce the computation time needed to perform multi-millenial experiments. However, accelerated experiments cannot correctly represent the effect of freshwater discharge to the ocean resulting from ice sheet melting since either the flux of water or the mass can be preserved, but not both at the same time. Here, we discard completely the role of freshwater flux to the ocean in the accelerated experiments. The ADGL experiments is 270 the accelerated counterpart of the DGL experiments, and as such will define the new reference for the accelerated experiments.

The other accelerated experiments are used to assess the sensitivity of our simulated deglaciation to important processes related to ice sheet dynamics: modelling choices for ice dynamics, for the surface mass balance and for the sub-shelf melt rate.

First, we explore two aspects related to ice dynamics: grounding line dynamics and ice deformation. The ice sheet model 275 GRISLI accounts for two formulations of the flux at the grounding line. For the Antarctic ice sheet, the use of Schoof (2007) instead of Tsai et al. (2015) leads to slower grounding line retreat during deglaciation phases (Quiquet et al., 2018a). For this reason, in the ADGL_schoof experiment we use the Schoof (2007) formulation of the flux at the grounding instead of Tsai et al. (2015). A second aspect for ice dynamics is the choice of the flow enhancement factor $E_f$ which is a tuned parameter that has consequences on the ice velocity. In the ADGL_ef experiment we use a larger flow enhancement factor (larger velocities) 280 since the simulated North American ice thickness at the LGM is overestimated (Fig. 2).

Then, to explore the sensitivity of our results to the surface mass balance we have performed two experiments in which the weakly constrained melt parameter $c_{rad}$ (Eq. 3) is changed. In ADGL_accplus we use a smaller value for this parameter in order to reduce surface melt to delay the deglaciation. In the ADGL_nocor experiment we use an homogeneous value of $c_{rad}$ instead of using the spatial heterogeneous value defined from the temperature bias.

285 Finally, to assess the sensitivity of our results to the sub-shelf melt rate, in the ADGL_bmbplus we enhance the sub-shelf melt rate to increase the relative importance of oceanic changes with respect to atmospheric changes.

The list of the different experiments is available in Tab. 1.





**Table 1.** Characteristics of the experiments performed in this study. The experiments marked with * use a reduction coefficient for the freshwater flux to the ocean resulting from ice sheet melting.

| Label | Accelerated | Freshwater | Brines | Grounding line flux | $E_f$ (-) | $c_{rad}$ (W m$^{-2}$) | $F_g$ (-) |
|---|---|---|---|---|---|---|---|
| DGL | No | Yes | No | Tsai et al. (2015) | 1.8 | variable, -40 | $15\times10^{-3}$ |
| DGL_FWF/2 | No | Some* | No | Tsai et al. (2015) | 1.8 | variable, -40 | $15\times10^{-3}$ |
| DGL_FWF/3 | No | Some* | No | Tsai et al. (2015) | 1.8 | variable, -40 | $15\times10^{-3}$ |
| DGL_noFWF | No | No | No | Tsai et al. (2015) | 1.8 | variable, -40 | $15\times10^{-3}$ |
| DGL_brines | No | Yes | Yes | Tsai et al. (2015) | 1.8 | variable, -40 | $15\times10^{-3}$ |
| ADGL | Yes | No | No | Tsai et al. (2015) | 1.8 | variable, -40 | $15\times10^{-3}$ |
| ADGL_schoof | Yes | No | No | Schoof (2007) | 1.8 | variable, -40 | $15\times10^{-3}$ |
| ADGL_ef | Yes | No | No | Tsai et al. (2015) | 3.5 | variable, -40 | $15\times10^{-3}$ |
| ADGL_accplus | Yes | No | No | Tsai et al. (2015) | 1.8 | variable, -50 | $15\times10^{-3}$ |
| ADGL_bmbplus | Yes | No | No | Tsai et al. (2015) | 1.8 | variable, -40 | $150\times10^{-3}$ |
| ADGL_nocor | Yes | No | No | Tsai et al. (2015) | 1.8 | homogeneous, -40 | $15\times10^{-3}$ |

## 3 Results

In this section we first describe the general evolution of the simulated climate in the synchronously coupled experiments before examining the ice sheet changes. Then we examine the results for the accelerated asynchronously coupled experiments to infer the sensitivity of our results to different ice sheet evolutions.

### 3.1 Climate evolution in the synchronously coupled ice sheet – climate experiments

The simulated global mean surface temperature evolution for the synchronously coupled ice sheet climate experiments is shown in Fig. 3, together with the strength of the AMOC. In response to the forcings, the different experiments produce a gradual warming from the last glacial maximum towards its maximum value during the Holocene. The glacial-interglacial temperature difference ranges from 3.1 to 3.8°C and is in good agreement with palaeo-temperature stack (Shakun et al., 2012), even though iLOVECLIM is one of the warmest model at the LGM within the PMIP4 ensemble (Kageyama et al., 2020) . The glacial-interglacial temperature difference is mostly explained by the cold temperatures at the LGM resulting from the large ice sheets that induce higher surface elevations and a strong albedo effect. A polar amplification is simulated since the northern and southern high latitudes both show a greater temperature difference from the pre-industrial compared to the Tropics (Fig. 4). This pattern is consistent with recent reconstructions (e.g. Tierney et al., 2020, shown in Fig. 4d), even though with a smaller amplitude in our model. However, our simulated glacial-interglacial temperature difference is within the range of other estimates (4 ± 0.8°C, Annan and Hargreaves, 2013).



For all the experiments, we simulate a gradual warming with no abrupt climate transitions. If the different experiments show a similar temperature evolution, they also display subtle differences. First, the experiments that use a reduced freshwater flux resulting from ice sheet melting present a more rapid warming compared to the reference experiment (e.g. DGL_noFWF with

respect to DGL). Second, the experiments show a diverging temperature evolution after around 13 kaBP. After this date, the reference DGL simulation shows a slight decrease in temperature for about 2 kyrs followed by a moderate warming until ~7 kaBP. On the contrary, the experiment in which the freshwater flux are discarded (DGL_noFWF) displays a brief period during which the temperature ceases to increase followed by a sharp temperature increase. In this case, the maximal surface temperature is reached at 10 kaBP after which there is a slight decrease until 7 kaBP. DGL_FWF/3 shows a very similar temperature change

as DGL_noFWF while DGL_FWF/2 presents similarities with both DGL and DGL_noFWF. This temperature evolution is in overall agreement with the temperature reconstruction of Shakun et al. (2012) which shows a pause in the deglacial warming trends at about 13.5 kaBP, synchronous with the carbon dioxide plateau. The experiment with the parametrisation of brines sinking, DGL_brines, displays a comparable temperature evolution to the reference simulation DGL for most of the simulated time period. However, the brine parametrisation induces a cooling of about 0.5°C in the first years after its activation due

to increased sea-ice extent around Antarctica. In addition, at 4 kaBP, the global mean temperature starts to rise again after a relatively steady state for the rest of the Holocene. At 0 kaBP the temperature in the DGL_brines experiment is close to the temperature in the DGL_noFWF and DGL_FWF/3.

These differences in terms of global mean surface temperature amongst the different experiments are mostly explained by the

differences in the state of the simulated Atlantic oceanic circulation. The reference experiment DGL simulates a decrease in the AMOC from the last glacial maximum. After a 50% reduction in its glacial values, the oceanic circulation strengthens at 13.5 kaBP for about 500 years before an abrupt collapse. This AMOC collapse is synchronous with the simulated pause in the temperature increase. From 12 kaBP onwards, the model simulates virtually no meridional overturning circulation. The evolution of the AMOC is drastically different when the freshwater flux to the ocean resulting from ice sheet melting is not

considered (DGL_noFWF). In this case, the AMOC remains strong during the whole 26 ka, with a maximum in the middle of the deglaciation towards 14 kaBP. This explains why the temperature rises more rapidly during the deglaciation in this experiment compared to the reference DGL experiment. Due to the weak AMOC in this case, the Northern Hemisphere remains colder, which ultimately delays the deglaciation of the ice sheets. The simulated pre-industrial period is also 0.8 °C colder in DGL with respect to DGL_noFWF since the absence of oceanic meridional heat transport results in much colder high lati-

tudes, especially in the North Atlantic (Fig. 5). The release of only half the meltwater flux to the ocean (DGL_FWF/2) does not allow to maintain an active AMOC during the Holocene neither but the collapse of the AMOC is delayed here with respect to the reference experiment. In addition to the DGL_noFWF experiment, only the experiment in which only one third of the meltwater flux is released to the ocean (DGL_FWF/3) is able to maintain an active AMOC during the Holocene. In this case, there are several abrupt oscillations in the strength of the circulation from 14 to 10 kaBP, but the model recovers and simulates

an AMOC similar to the DGL_noFWF from 10 kaBP onwards. For most of the simulated time period, the experiment in which the sinking of brines around Antarctica is parametrised (DGL_brines) shows a very similar evolution than the reference DGL



experiment, except that the AMOC shut-down occurs a few centuries earlier. However, at 4 kaBP the AMOC abruptly recovers and explain the final increase in the global mean temperature.

While some experiments show very abrupt shifts in the ocean, the atmospheric temperature evolution is nonetheless mostly gradual. This is visible at the global scale (Fig. 3b), but also when examining the temperature change above the Greenland ice sheet (Fig. 6a). The local temperature change closely resembles the global mean temperature change, even though with a larger amplitude. There are a few abrupt changes : a bit less than 4°C in about 200 years at 10.7 kaBP and at 3.8 kaBP for the DGL_FWF/3 and DGL_brines experiments, respectively. These are direct consequences of the AMOC recoveries visible

in Fig. 3c. These simulated abrupt warming events over the Greenland ice sheet look similar to the ones of the ice core record (Fig. 6b). The North GRIP $\delta^{18}$O is generally used to reconstruct the past local temperature changes with a conversion factor of 0.67 to 0.8 ‰ per degree (e.g. Johnsen et al., 1997; Buizert et al., 2014), suggesting a glacial-interglacial difference of more than 15°C. On comparable timescales, the Bølling-Allerød warming at 14.7 kaBP displays a similar temperature change amplitude compared to our simulated abrupt warming events, even though slightly larger. This suggests that, in our model, abrupt

changes of the Atlantic oceanic circulation can induce large temperature changes over the Greenland ice sheet, similar to the ones deduced from the ice core records. However, the timing of the simulated abrupt events in the experiments shown here does not correspond to the ones of the ice record.

## 3.2   Simulated ice sheet changes

The large-scale differences amongst the different experiments discussed in Sect. 3.1 are largely driven by differences in the amount of the freshwater released to the ocean related to ice sheet melting. This freshwater flux is shown in Fig. 7a for the reference experiment DGL. Even though this flux displays some variability, its evolution is generally gradual and shows a maximum around 14 kaBP where it peaks above 0.3 Sv (1 Sv corresponds to $10^6$ m³/s) with 100-yr mean values about 0.23 Sv. In Fig. 7a we also show the meltwater flux computed from the ice thickness changes in the ICE-6G_C and GLAC-1D geologi-

cally constrained reconstructions. These fluxes have the same order of magnitude of the simulated flux in the DGL experiment. However, the model fails to reproduce the two distinct accelerations in ice sheet retreat visible in the reconstructions for the meltwater pulse 1A at 14.6 kaBP (Deschamps et al., 2012) and the meltwater pulse 1B at 11.45 kaBP (Abdul et al., 2016). Instead, the model produces important fluxes (greater than 0.1 Sv) over a few thousand years. An other way to discuss these fluxes is to integrate them in time to have an idea of the total ice sheet volume evolution through the deglaciation (Fig. 7b). In

doing so, we can see that the coupled iLOVECLIM-GRISLI model setup produces an ice volume evolution in general agreement with the reconstructions since it lies most of the time between the two estimates of ICE-6G_C and GLAC-1D. However, the coupled model seems to deglaciate too fast since it displays a lower total ice sheet volume than the two reconstructions from 12.5 kaBP. In Fig. 7b we also show the eustatic sea level reconstruction of Lambeck et al. (2014) which displays a larger ice sheet volume, in particular around the last glacial maximum. Since we do not simulate the Antarctic ice sheet changes,

the ice volume shown in this figure only represents the Northern Hemisphere ice sheet volume. Interactive simulation of the





Antarctic ice sheet would result in a larger ice volume during the glacial period reducing partially the mismatch with the Lambeck et al. (2014) reconstruction. At the end of the simulation, the model has an overestimation of the present-day ice volume. This overestimation corresponds to about 4.5 m of sea level equivalent and is explained by an overestimation of the Greenland ice sheet volume and remaining small ice sheets in the Ellesmere Islands, Iceland, Norway and off-coast of Newfoundland.


The ice volume evolution of individual ice sheets is presented in Fig 8 for both the reference DGL and the DGL_noFWF experiments. In this figure, the individual ice sheet break-up is also represented for the ICE-6G_C and the GLAC-1D reconstructions. The ice volume partitioning is well reproduced. The North American ice sheet is by far the largest contributor for the last glacial sea level fall. At 26 kaBP, we simulate an ice volume of 81 m of sea level equivalent within the range of the
geologically reconstructions (75 and 86 m). However, in our experiments, the North American ice sheet volume increases until 20.5 kaBP where the reconstructions suggest a decline already as early as 26 kaBP (ICE-6G_C) or 23.8 kaBP (GLAC-1D). This is mostly due to our methodology used to define the initial state for the coupled experiments. When the coupling starts, at the beginning of our experiments, there is an abrupt change in the climate model in terms of ice mask and surface elevation, from GLAC-1D to our spun-up ice sheets. Our spun-up ice sheets at 26 kaBP (Fig. 2a) show a higher North American ice
sheet surface elevation than the GLAC-1D reconstruction used during the climatic spin-up, suggesting an overestimation of the precipitation in this area. When the coupling starts, this precipitation bias is amplified due to higher surface elevation and related increased orographic precipitation. The iLOVECLIM climate model likely shows an underestimation of the elevation desertification effect over the ice sheets (Quiquet et al., 2018b). The simulated volume of the Eurasian ice sheet displays a similar evolution than the North American ice sheet with a maximum around the last glacial maximum. This agrees well with
the GLAC-1D reconstruction. Given its smaller volume, the absolute rate of volume loss is smaller for the Eurasian ice sheet (1.3 m per millenia) compared to the one of the North American ice sheet (5.7 m per millenia). However, the Eurasian ice sheet already lost half its volume by 14.5 kaBP where this occurs at 12.8 kaBP for the North American ice sheet. The Greenland ice sheet presents only a small volume reduction of 2.6 m of sea level equivalent, in good agreement with the reconstructions. However, the Greenland ice sheet volume at the end of the simulation is largely overestimated compared to the present-day
observations (about 40% volume overestimation). As for the total volume, the individual ice sheets deglaciate faster in the DGL_noFWF experiment. This is particularly visible for the North American ice sheet for which there is a difference of one thousand years at about 11 kaBP.

A map of the simulated ice sheet configuration for selected snapshots is shown in Fig. 9. This figure shows the results for the
reference DGL experiment while the other synchronously coupled experiments show a similar deglacial pattern although with differences in timing. At 26 kaBP, the North American ice sheet presents on its northern margin some very active ice streams, from East to West: the Hudson Strait ice stream, the Lancaster Sound ice stream and the Amundsen Gulf ice stream. In these regions, grounded ice velocities are greater than 500 m/yr. Elsewhere, the ice sheet does not present well-identified ice streams but the margins present generally large velocities, greater than 200 m/yr. The other ice sheets present a smaller ice flow. From
26 to 21 kaBP, there is only little change of the ice sheet except the Eurasian ice sheet retreat from the British Isles and the





development of an ice shelf at the outlet of the Hudson Strait ice stream. The simulated topography at 21 kaBP (Fig. 2b) is close to the spun-up ice sheets used at 26 kaBP and remains generally in good agreement with the geologically constrained reconstructions. From 21 kaBP, we simulate a gradual ice sheet retreat for both the North American and the Eurasian ice sheets. The North American ice sheet mostly retreats in its southern continental part due to decrease surface mass balance

related to the gradual warming. The deflected bedrock in this area leads to the apparition of proglacial lakes, already visible at 14 kaBP. Similarly, at this date, the southern flank of the Eurasian ice sheet also displays proglacial lakes. The eastern part of the Eurasian ice sheet, the Barents-Kara ice sheet, rapidly collapses due to a grounding line instability in the Kara sea. This instability is initiated at about 14.5 kaBP and results in a complete desintegration of the Barents-Kara ice sheet in about 1.2 ka. An other grounding line instability occurs later for the continental part of the North American ice sheet. The grounding

line retreat is clearly visible at 12 kaBP. This lake-induced instability considerably facilitates the North American ice sheet deglaciation. At 8 kaBP, we simulate a very small North American ice sheet and only a relic of the Eurasian ice sheet over the Scandinavian mountains. At this time, the bedrock is still depressed below sea level over the northern most part of America but slowly returns to its present-day value. During the last 1000 years of the simulation, the bedrock uplift rate in the vicinity of the Hudson Bay is about 0.5 m to 1.2 m per century, a value comparable to modern observations (Husson et al., 2018). The

Greenland ice sheet expands considerably onto the continental shelf during the glacial period and retreats until about 10 kaBP. It does not display any substantial change in the ice extent during the Holocene but it displays some ice elevation changes. The ice elevation evolution near the summit shows a maximum at about 10 kaBP and decreases afterwards in agreement with palaeo-elevation reconstructions at the deep ice core drilling sites (Vinther et al., 2009).

In Fig. 10 we present the rate of total ice mass change and its individual components: surface mass balance, basal mass balance and calving. The total mass change remains positive until 20.5 kaBP due to a positive integrated surface mass balance, not entirely compensated by the basal mass loss (mostly sub-shelf melt) and calving. After this date, the total mass change becomes negative for the rest of the duration of the experiment. The total mass loss peaks at -9.7 $\times$ 10$^3$Gt yr$^{-1}$ at 13.8 kaBP when surface ablation and loss by calving almost synchronously display a maximum (surface ablation slightly precedes the calving

increase). At this date, the mass loss due to the ocean and lake represent more than half the loss by surface mass balance. In fact, if both basal mass loss and calving remain almost constant until 14.5 kaBP (-1.4 $\times$ 10$^3$Gt yr$^{-1}$), they nonetheless show some variability after this date. These fluxes are maximal at the time of the grounding instabilities shown in Fig. 9 for both the Eurasian (14.5-13.5 kaBP) and the North American (12.8-10 kaBP) ice sheets. While the mass loss is primarily driven by surface ablation until 12.8 kaBP, after this date the oceanic and lake forcing become the major driver for the ice sheet retreat.

The total mass loss finally reaches zero (ice sheet equilibrium) at 6.5 kaBP.

### 3.3 Accelerated experiments to assess specific sensitivities

The aim of this section is to assess the sensitivity of the simulated climate evolution to the choice of critical ice sheet model parameters and assumptions. To do so, we have performed additional experiments in which the forcings are accelerated. Three





major sources of uncertainties have been explored: ice sheet mechanics (deformation and grounding line dynamics), surface mass balance and sub-shelf melting rates.

The evolution of some large-scale climate variables for these additional experiments are shown in Fig. 11. Since we do not feed back the freshwater related to ice sheet melting to the ocean in the accelerated experiments, they have to be compared to the

DGL_noFWF experiment. The reference accelerated experiment ADGL (black in Fig. 11) is in fact similar to the DGL_noFWF (grey): rapid temperature increase and active Atlantic circulation throughout the deglaciation. However, the accelerated experiment displays larger ice sheets than the non-accelerated (about 8 m of sea level equivalent at 14 kaBP) and as a result a colder climate (~0.4°C in global mean surface temperature at 14 kaBP). If the timing of the ice sheet retreat can be slightly different, the overall pattern of this retreat is only weakly affected by the acceleration factor.


The two experiments related to ice sheet dynamics (ADGL_ef and ADGL_schoof) do present some differences in their simulated ice sheet volume. The increased enhancement factor (ADGL_ef) leads to thinner ice sheets (smaller ice volume) and, as such, deglaciates faster than the reference accelerated experiment (ADGL). The experiment in which we use the formulation of Schoof (2007) instead of Tsai et al. (2015) (ADGL_schoof) also produces a lower ice sheet volume during the glacial period.

However, this experiment shows a slower ice sheet retreat during the deglaciation compared to the reference ADGL experiment. This is mostly related to the greater grounding line sensitivity in the formulation of Tsai et al. (2015), already shown in (Quiquet et al., 2018a) for the Antarctic ice sheet. These differences in terms of ice sheet evolution have nonetheless only a limited impact on the climate evolution. The ADGL_ef produces a slightly more rapid warming during the deglaciation (related to the smaller ice sheets) while it is the contrary for the ADGL_schoof (slower ice sheet retreat). The Atlantic circulation is also

weakly impacted by the different ice sheet evolution. Only the ADGL_ef produces a decrease in the overturning circulation strength slightly earlier than the ADGL experiment while the ADGL_schoof displays insignificant differences.

The two experiments related to modification of the surface mass balance parameters induce larger simulated ice sheet volume differences. Associated with a larger ice sheet surface mass balance, the ADGL_accplus produces larger ice sheet volumes

throughout the whole simulated time period. For this experiment the maximal ice volume is reached circa 19 kaBP and it is larger by about 15 m of sea level equivalent than the ADGL experiment. This excess ice also explains the delayed ice sheet retreat: at 10 kaBP the simulated ice sheets still represent about 45 m drop in eustatic sea level in the ADGL_accplus experiment compared to about 8 m in the ADGL experiment. This has consequences on the simulated climate: i- the global mean temperature rises more slowly, reaching eventually a comparable value to the reference simulation at 0 kaBP; ii- the phase of

very active overturning in the middle of the deglaciation is extended by 2 ka. Even though the ADGL_accplus experiment displays larger ice sheet volume, the pattern of the ice sheet retreat is similar to the one of the ADGL experiment. On the contrary, the ADGL_nocor experiment provides alternative ice sheet histories. In the ADGL experiment the Barents-Kara sector of the Eurasian ice sheet is almost fully deglaciated at 13.5 kaBP, while it is the case only after 8.5 kaBP in the ADGL_nocor experiment. Conversely, the North American ice sheet retreats faster in the ADGL_nocor experiment. This is a direct consequence



of the cold temperature bias in Northern Europe and the warm bias in North America. If the climate evolution is not drastically changed as a result of these different ice sheet chronologies, it nonetheless shows some interesting differences. The overturning circulation remains moderate for a longer time period compared to the ADGL experiment since it increases only after 15 kaBP (w.r.t. 17.5 kaBP in ADGL). As a result the global mean temperature in ADGL_nocor is colder than in ADGL even though it shows smaller ice sheets at least until 15 kaBP. The oceanic circulation in the model seems largely affected by the Eurasian ice

sheet size.

Finally, the experiment in which we increase the sub-shelf melting rate, ADGL_bmbplus, shows only negligible changes with respect to the ADGL experiment. This suggests that, in our model, the ice sheet retreat is mostly driven by surface ablation and not sub-shelf melt.


## 4 Discussion

We have shown that in our reference experiment the freshwater flux to the ocean resulting from ice sheet melting leads to a progressive weakening of the Atlantic overturning circulation from the last glacial maximum, eventually leading to a complete shut-down without recovery within the time frame of the experiments presented. With different sensitivity experiments

in which we modify the amount of freshwater flux released to the ocean, we have shown that we are able to simulate abrupt transitions from collapsed to recovered state of the Atlantic circulation during the deglaciation. Thus, with a reduced freshwater flux, the AMOC can remain active during the Holocene. This suggests that if the model contains the physical elements for rapid changes of the AMOC, it seems nonetheless too sensitive to the amount of freshwater since it is unable to maintain an active oceanic circulation with realistic amount of freshwater fluxes. Alternative experiments (not discussed here) with the iLOVE-

CLIM model in which we used prescribed ice sheet reconstructions (instead of interactive) and freshwater fluxes derived from the GLAC-1D and ICE-6G_C also lead to a shut-down of the overturning circulation. This problem has been identified in other models. For example, freshwater derived from geologically-constrained ice sheet reconstructions (ICE-5G, Peltier, 2004) also lead to an AMOC collapse in Bethke et al. (2012) while most of the time idealised freshwater scenarios, which can substantially differ from the reconstructions, are preferred (e.g. Liu et al., 2009; Menviel et al., 2011; He et al., 2013; Obase and Abe-Ouchi,

2019). Transient sensitivity of the simulated AMOC to freshwater flux remains an open question when attempting to simulate the climate evolution across the last deglaciation. For these transient experiments, it would be useful to perform a systematic analysis of the sensitivity of the oceanic circulation to key processes for deep convection, such as the brines rejection during sea ice formation or atmospheric wind stress, and also in the way the freshwater flux are imposed to the oceanic model, e.g. considering the depth of the freshwater release, its seasonality or the impact of the iceberg transport.


The simulated temperature change during the deglaciation is generally very gradual with no abrupt transitions. For example, in our experiments, over the Greenland ice sheet, the local temperature change is strongly correlated to the global mean tem-



perature change and most of the time does not display abrupt events such as the one recorded in ice cores (Alley, 2000b). In fact only the abrupt AMOC recoveries in certain experiments (DGL_FWF/3 at 10.7 kaBP and DGL_brines at 3.8 kaBP) are

able to produce abrupt temperature changes in Greenland comparable to the ice core record. Since these AMOC recoveries are lacking in the majority of our experiments we generally largely underestimate the millenial scale variability observed at high latitudes. This variability could largely influence the ice sheet evolution. For example, since the Bølling-Allerød warming is not simulated in our model, we are not able to quantify its impact on the North American or the Eurasian ice sheets (Gregoire et al., 2016; Brendryen et al., 2020).


In addition, within the experiments presented here, only changes in the AMOC related to freshwater flux are able to produce some abrupt temperature changes. For example, all the accelerated experiments, in which this process is not considered, produce a smooth temperature increase since the LGM. However, these experiments show different ice sheet evolutions with some rapid ice sheet retreat at times. This suggest that in our model and for the time period simulated, external forcing and ice sheet

changes alone are not able to produce millenial-scale climate variability without invoking freshwater hosing.

Finally, in our experiments we did not consider the potential changes of the Antarctic ice sheet since we use a constant topography and ice mask in the Southern Hemisphere. Similarly we do not take into account the freshwater flux resulting from Antarctic ice sheet retreat from the last glacial maximum. This simplification was motivated by the fact that an earlier study

already identified that freshwater hosing around Antarctica with our model has a negligible impact on the simulated climate (Roche et al., 2010). In this region, the circumpolar current tends to rapidly dilute the released freshwater leading to a very limited impact on vertical oceanic mixing. However, the gradual retreat of the ice sheet from the continental shelf margin can also facilitate the sinking of brines and as such enhance dense water formation. If the sinking of brines around Antarctica seems to play a moderate role in our experiments, it can nonetheless produce an abrupt AMOC recovery at 3.8 kaBP, not occurring in

the reference experiment. As such, this process should be more thoroughly investigated with, e.g., interactive Antarctic topography and bathymetry.

## 5   Conclusions

In this paper, we have presented climate model experiments in which the Northern Hemisphere ice sheets are synchronously

coupled to the rest of the system (atmosphere and ocean). For the majority of our experiments, the atmospheric changes are mostly gradual while the Atlantic overturning circulation displays abrupt changes. In the reference experiment, the model fails at keeping an active circulation during the Holocene. It is only when the freshwater amounts release to the ocean are reduced that we can simulate AMOC shut-downs and recoveries, suggesting a too strong sensitivity of this process in our model. The AMOC recoveries, when simulated, are associated with abrupt warming events in Greenland. The simulated ice

sheet evolution is in general agreement with geologically reconstructions even though the retreat is too fast with respect to

these reconstructions. The simulated ice sheets present some phases of acceleration in their retreat related to grounding line instabilities. These occurs in the Arctic ocean for the Eurasian ice sheet and in proglacial lakes at the southern margin of the North American ice sheet. However these events are not directly correlated to abrupt climate changes. In addition, we performed various sensitivity experiments in which we did not consider the freshwater released to the ocean but in which we

modified some critical aspects of the ice sheet model. If these experiments produce different ice sheet deglacial chronologies they show similar climate trajectories. This suggests that ice sheet geometry changes alone, i.e without freshwater fluxes, are not enough to generate abrupt events in our model.

*Data availability.* Archiving of source data of the figures presented in the main text of the manuscript is underway. Data will be made publicly available upon publication of the manuscript on the Zenodo repository with digital object identifier 10.xxxx/zenodo.xxxxxxx. They

are temporarily available for review purposes upon request.

*Author contributions.* A.Q., D.M.R., C.D. designed the project. All authors have contributed to the model developments necessary to perform this work. A.Q. performed the simulations. All authors participated in the analysis of model outputs and the manuscript writing.

*Competing interests.* The authors declare no competing interests.

*Acknowledgements.* C. Ritz is warmly thanked for her implication on the GRISLI ice sheet model. We acknowledge the Institut Pierre Simon

Laplace for hosting the iLOVECLIM model code under the LUDUS framework project (https://forge.ipsl.jussieu.fr/ludus).



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





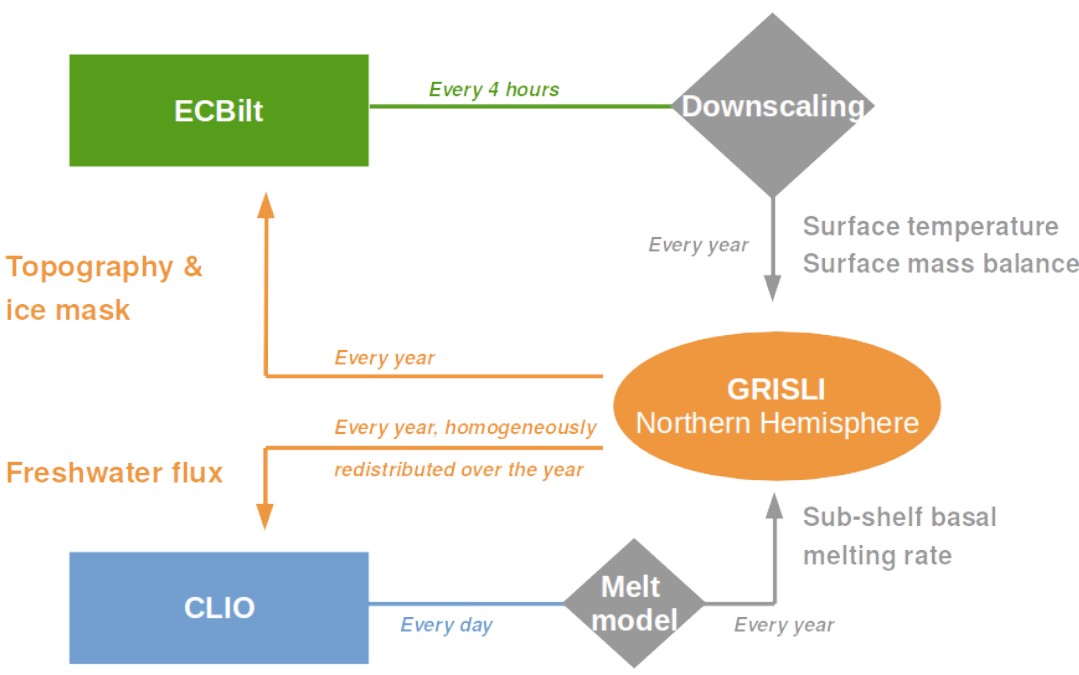

**Figure 1.** Schematic representation of the coupling between the ice sheet model (GRISLI) and the atmospheric (ECBilt) and the oceanic (CLIO) models.





**Figure 2.** Surface elevation above contemporaneous sea level: **(a)** after the glacial spin-up, **(b)** in the reference deglaciation experiment DGL at 21 kaBP, **(c)** in the GLAC-1D reconstruction and **(d)** in the ICE-6G_C reconstruction. The colour scale is different for ice-free and ice-covered regions. The simulated ice sheet grounding line is represented by the red line while the black lines represent isocontours of ice sheet surface elevation (separated by 1000 metres).



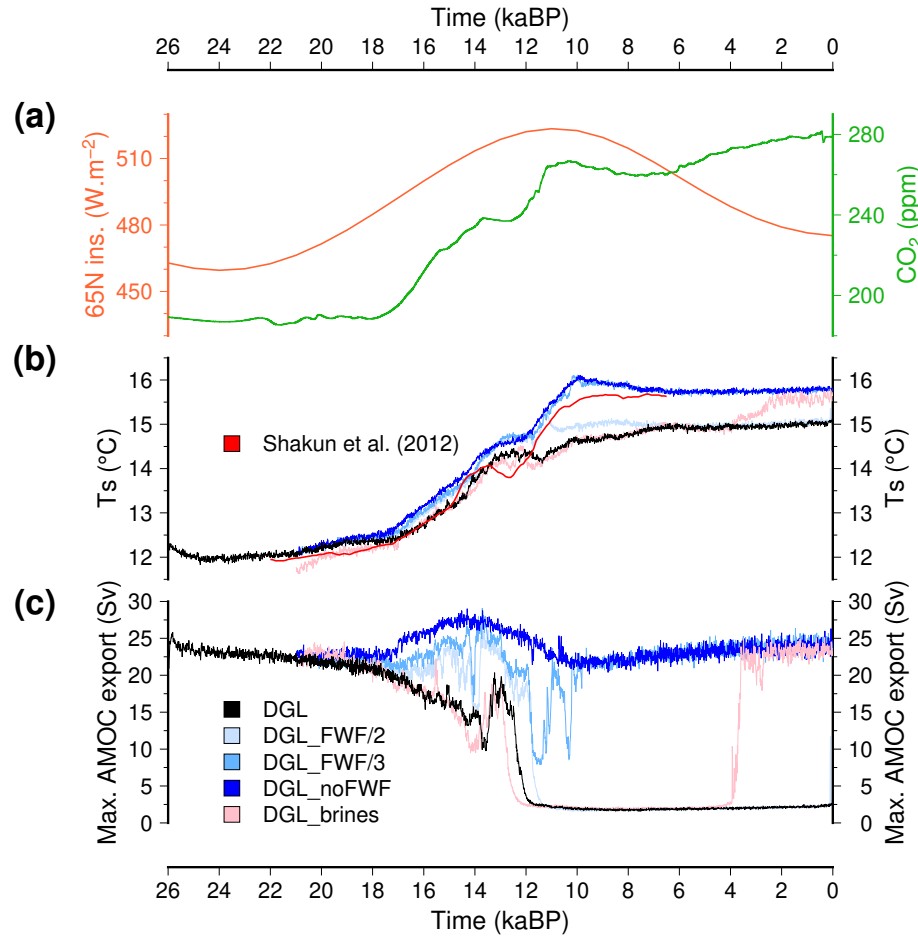

**Figure 3. (a)**: Time evolution of the major forcings for the climate model (June insolation at 65 °North and carbon dioxide mixing ratio). **(b)**: Simulated global mean surface temperature. **(c)**: Simulated maximum of the Atlantic stream function. The reference model DGL is in black while the experiments with reduced freshwater flux to the ocean from ice sheet melting are depicted with a blue shading (dark blue for no freshwater flux). The experiments with enhanced brine formation is in pink. Here, we use a 10-yr running mean for the model results to smooth interannual variability. In **(b)** we also show the temperature anomaly reconstruction from Shakun et al. (2012) (on which we added 15.5°C, a typical pre-industrial global mean surface temperature simulated by the model).





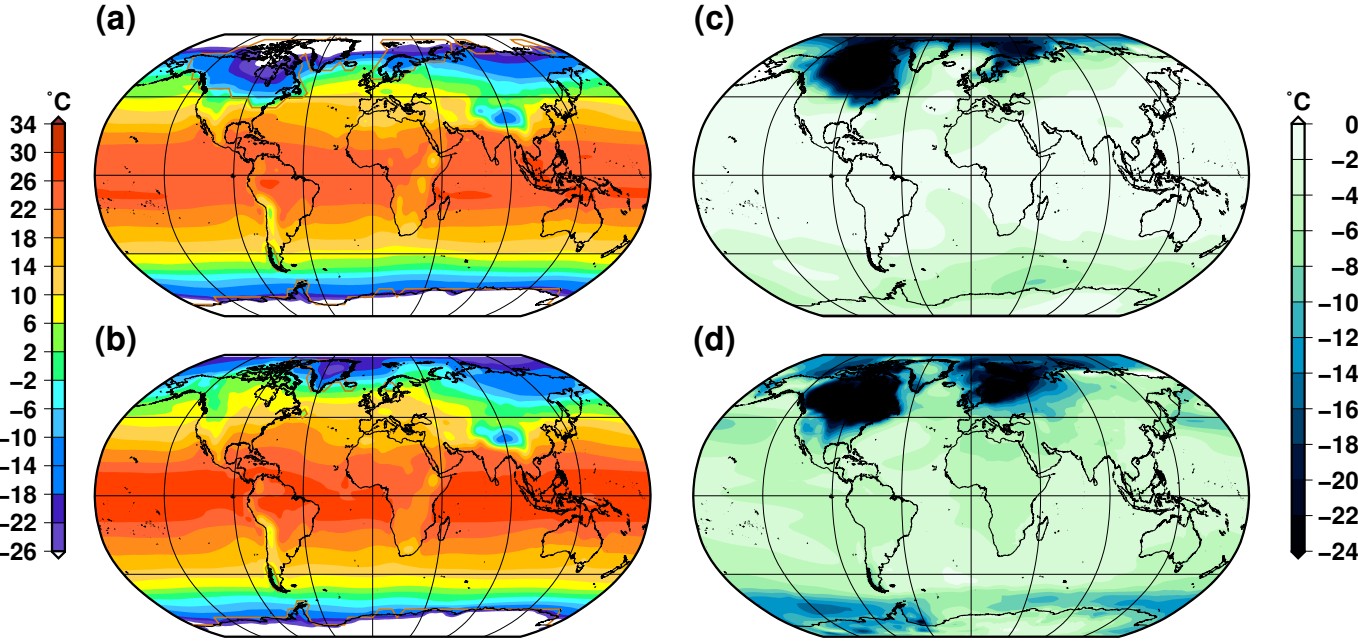

**Figure 4.** Simulated annual near-surface air temperature in the reference experiment DGL, **(a)**: at the last glacial maximum (21 ka BP) and **(b)**: for the pre-industrial (0 ka BP). **(c)**: Simulated temperature difference between the last glacial maximum and the pre-industrial (b-a). **(d)**: Temperature difference between the last glacial maximum and the pre-industrial in Tierney et al. (2020).

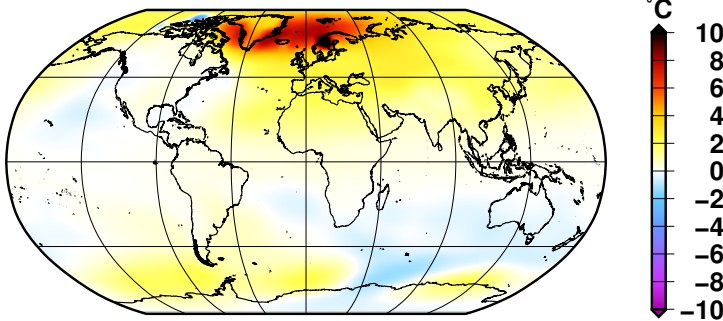

**Figure 5.** Simulated annual near-surface air temperature difference during the pre-industrial (0 ka BP) from the reference experiment DGL and the experiment DGL_noFWF in which the freshwater flux resulting from ice sheet melting is not applied to the ocean model (DGL_noFWF - DGL).



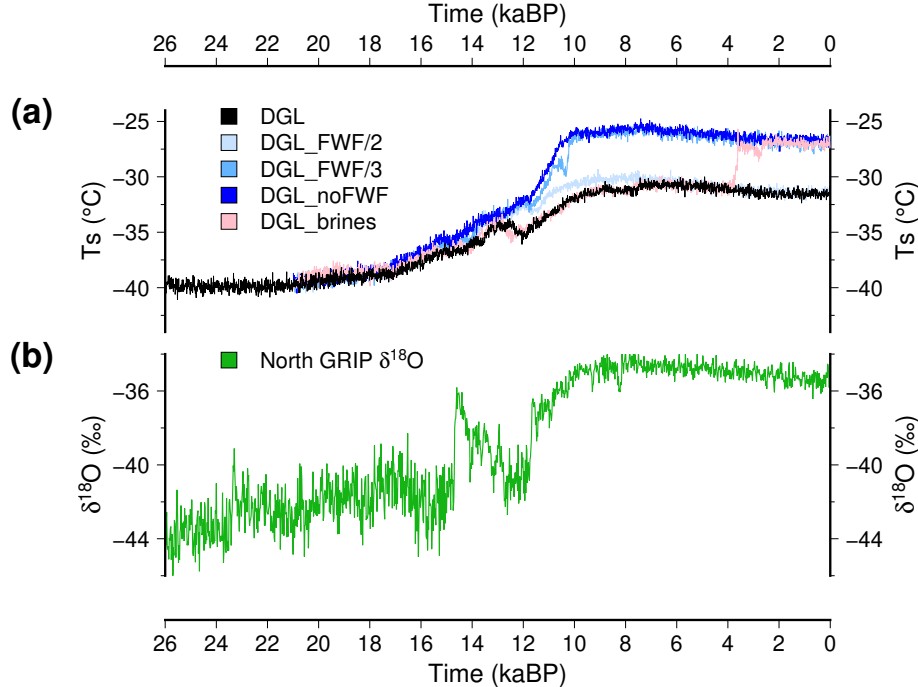

**Figure 6. (a)**: Simulated surface temperature at the location of the North GRIP deep ice core. The reference model DGL is in black while the experiments with reduced freshwater flux to the ocean from ice sheet melting are depicted with a blue shading (dark blue for no freshwater flux). The experiments with enhanced brine formation is in pink. Here, we use a 10-yr running mean for the model results to smooth interannual variability. **(b)**: The isotopic content in $\delta^{18}$O measured at North GRIP (Andersen et al., 2004), which is often considered as representative of local temperature changes.



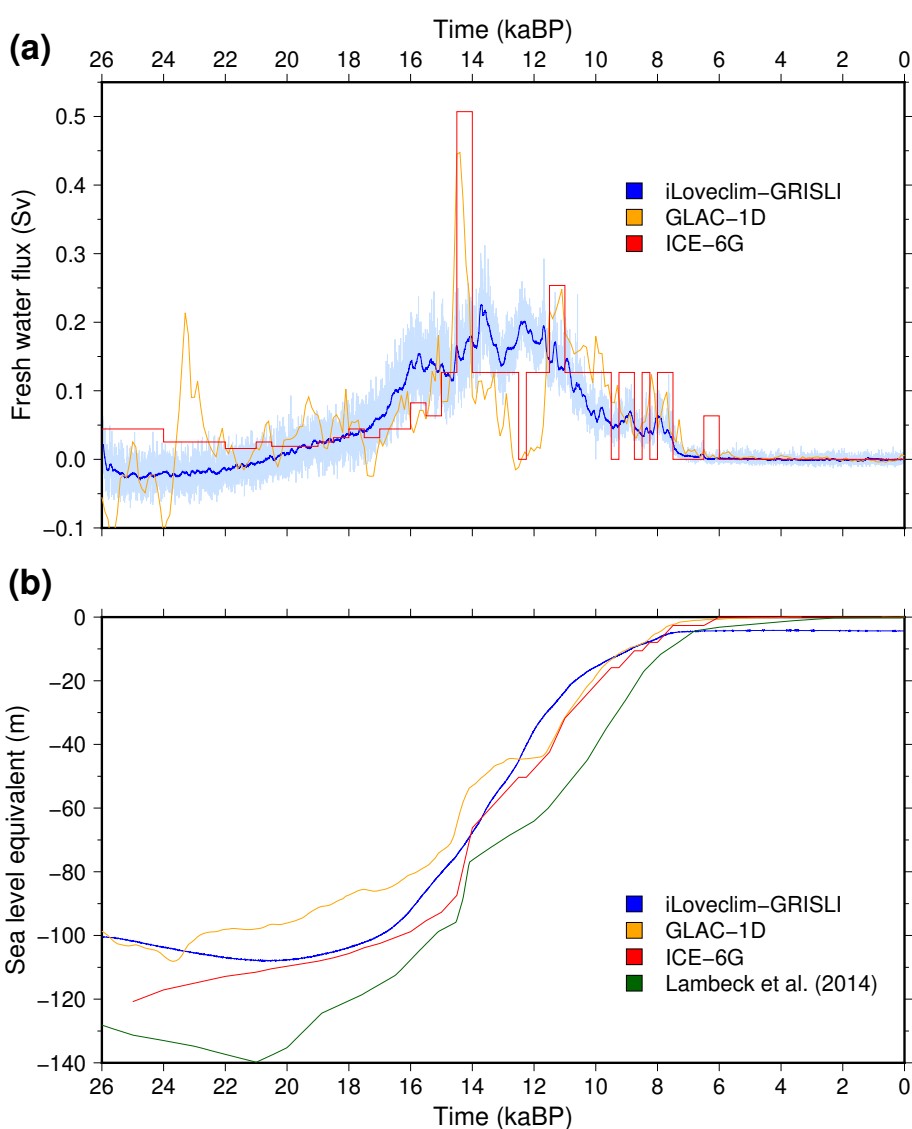

**Figure 7. (a)**: Time evolution of the freshwater release to the ocean resulting from the computed change in the North Hemisphere ice sheets. The blue curve depicts the values smoothed with a 100-yr running mean while annual values are depicted in light blue. The ice mass change for the two geologically-constrained reconstructions of GLAC-1D and ICE-6G_C are depicted in orange and red, respectively. **(b)**: Corresponding eustatic sea level evolution.



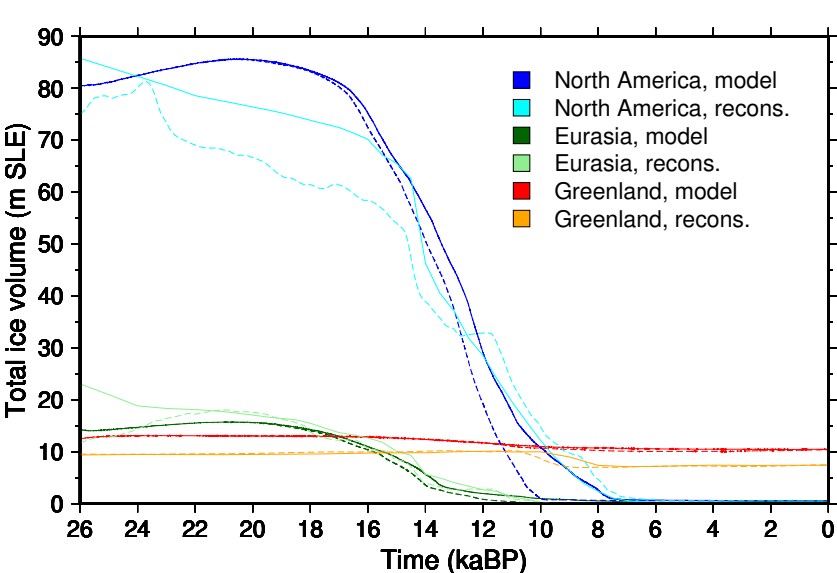

**Figure 8.** Individual ice sheet contributions to deglacial sea level rise, expressed as metres of sea level equivalent (SLE). For our model experiments, we show the ice volume for the reference experiment DGL (plain lines) and the experiment in which the freshwater flux resulting from ice sheet melting is not released to the ocean DGL_noFWF (dashed lines). For the reconstructions, we show the ICE-6G_C (plain lines) and the GLAC-1D (dashed lines) reconstructions.



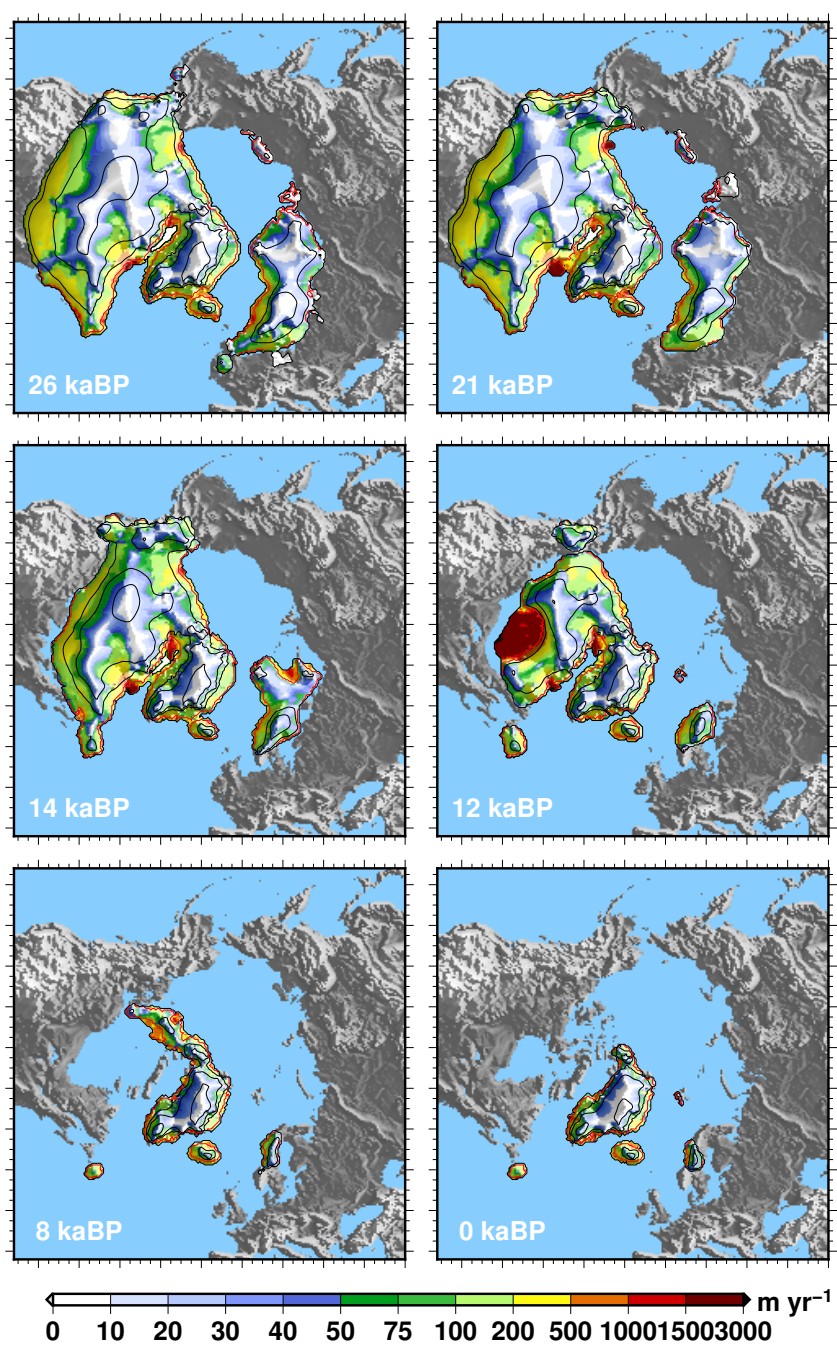

**Figure 9.** Simulated Northern Hemisphere ice sheets in the reference model for selected snapshots. The simulated ice elevation above contemporaneous eustatic sea level is shown with the black isocontours (separated by 1000 metres). The red contour is the ice sheet grounding line. The amplitude of the simulated vertically averaged ice sheet velocity is draped over the surface topography and depicted with the colour palette. Emerged land masses are in grey while bed elevation below contemporaneous sea level is in blue.





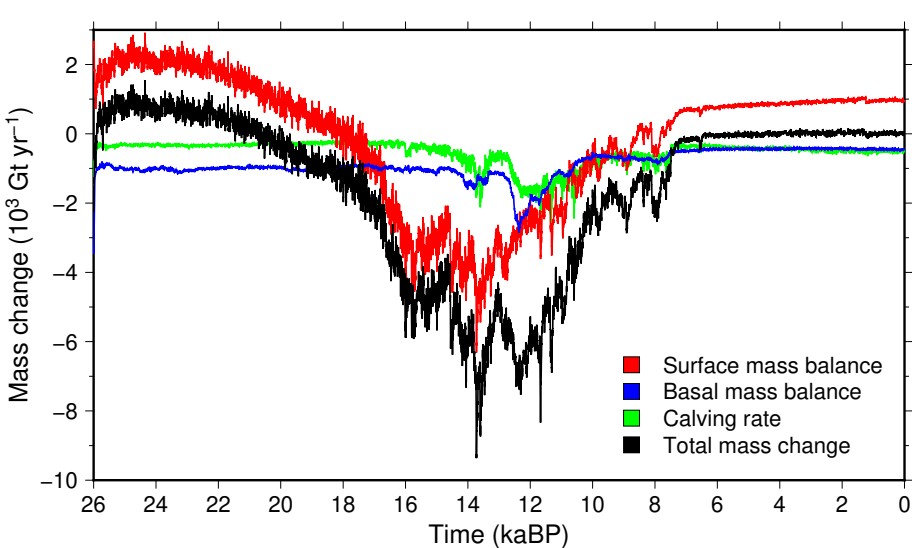

**Figure 10.** Time evolution of the different contributions to ice sheet mass changes. The total mass change is the sum of the surface mass balance, the basal mass balance and the calving rate.



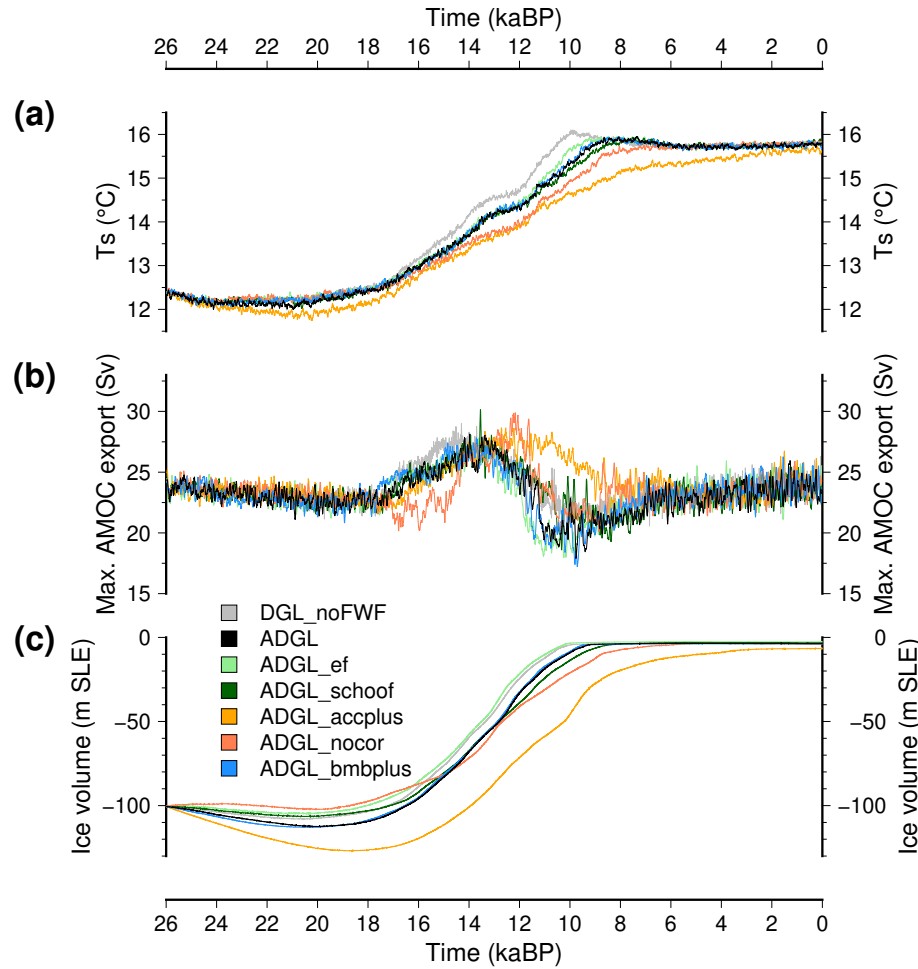

**Figure 11.** Time evolution of a selection of large-scale climate variables for different sensitivity experiments: **(a)** global mean surface temperature, **(b)** maximum of the Atlantic stream function and **(c)** simulated Northern Hemisphere ice volume. The model that does not account for the freshwater release to the ocean due to ice sheet melting is shown in grey (DGL_noFWF). The other lines are accelerated simulations (factor of acceleration of 5) and they similarly do not account for the freshwater flux to the ocean. The accelerated reference experiment is in blue (ADGL). The dark green line is in experiment in which we use the Schoof (2007) formulation of the flux at the grounding line (ADGL_schoof) instead of Tsai et al. (2015). The light green line is an experiment for which we use a larger enchancement factor (3.5 instead of 1.8, ADGL_ef). The light orange line is a version of the model with a lower $c_{rad}$ coefficient (-50 instead of -40 W m$^{-2}$, ADGL_accplus) which induces a more positive surface mass balance. The dark orange is for an experiment in which with do not apply the spatial correction of the $c_{rad}$ parameter (ADGL_nocor). The blue line is for an experiment with increase sub-shelf melting rate (ADGL_bmbplus).