# Peer review of "Climate and ice sheet evolutions from the last glacial maximum to the pre-industrial period with an ice sheet – climate coupled model"

_Climate of the Past, 2021_

## Referee Comment (RC1)

**Review of: Climate and ice sheet evolutions from the last glacial maximum to the pre-industrial period with an ice sheet – climate coupled model by Quiquet *et al**

Evan J. Gowan

`evan.gowan@awi.de`

**1   Overview**

Quiquet *et al.* present the results of a fully coupled ice sheet-climate model applied to the deglaciation of the Northern Hemisphere ice sheets from the Last Glacial Maximum. They use a climate model of intermediate complexity to couple in a bi-directional manner. Impressively, this is done at a very high frequency, with 5 years of ice sheet model for ever one year of simulated climate. The coupled model run produces a satisfactory representation of the deglaciation. One of the major findings is that the gradual input of freshwater in the ocean causes a shutdown of the AMOC. In some of their additional tests, they find that the AMOC does not shut down without this freshwater input. Other experiments showed only weak dependence on parameters related to surface and subglacial processes.

Overall, I enjoyed reading this paper, and found that it was well written. The experimental setup was well described (though note some of my comments below), and the implications for the ice sheet collapse on the overall climate was thorough. My comments below are mainly related to some parts of the modelling that should be elaborated a bit, and I do not think additional experiments are necessary. Therefore, I recommend relatively minor corrections.

**2   Comments**

**2.1   Computational overhead**

I am particularly impressed by 5/1 ratio of ice sheet to climate model years in this coupled setup. My question, though, is what is the computational costs of this model setup? If there is relatively low cost, then perhaps it could easily be adopted by other groups to investigate other problems, so this would be a good way to sell it.

**2.2   Ice sheet model resolution**

Perhaps as part of the last point, I am curious as to why an ice sheet model resolution of 40 km is used. Is this a computational limitation? In experiments done by our group here at the Alfred Wegener Institute, we found that the trajectory of the ice sheet evolution can be radically different just by going from 40 km to 20 km resolution. In general, the computational expense of ice sheet models tends to be pretty low compared to climate models, so I think some discussion on this choice needs to be made.

**2.3  Glacial isostatic adjustment**

The GIA model used for the experiments is not described, so I would ask that this be added. Looking at Figure 9, the topography is depressed far more than in reality, which causes broad glacial lakes to form along the southern margin that are much bigger than during the actual deglaciation. This is one of the possible reasons (or even the main reason) that deglaciation was faster than expected.

**2.4  Sediment thickness for basal sliding**

I don't understand why the Laske and Masters (1997) dataset was used to parameterize sediment distribution. That dataset is a map of Phanerozoic, undeformed sedimentary rock thickness for use in global seismology problems, and is vastly different to the distribution of unconsolidated sediments that would be important for ice sheet sliding. There are places with sedimentary bedrock where there are no unconsolidated sediments (for instance in Ontario and central Manitoba), and there are places on the Precambrian Shield where there are thick unconsolidated sediments (for instance the Thelon Basin in Kivalliq). As a first order approximation, I guess you could assume that areas where the bedrock is Phanerozoic sedimentary rocks are more likely to be covered by unconsolidated sediment, so I don't think it would radically alter the results. However, I suggest in the future that a different dataset be used. Full disclosure, I have created such a dataset for North America (Gowan et al., 2019).

**2.5  Spinup time**

A 200 kyr spinup is used to initialize the ice sheet to the LGM state. I'm wondering why such a long spinup was necessary, considering that during the last interglacial (about 100 kyr before the LGM), there were essentially no ice sheets in the Northern Hemisphere except for perhaps part of the Greenland Ice Sheet. Even the Eurasian Ice Sheets were probably almost non-existent just 15 kyr before the LGM (Hughes et al., 2016). Would such a long spinup affect the results?

**2.6  Comparison with geological data**

There is a section that compares the modeled results with some ice sheet reconstructions. I think this is fine, but don't feel too bad that you don't match things exactly, since the margin chronology in North America is in the process of being revised (Dalton et al., 2020). In some places the timing of advance and retreat is being revised by thousands of years. In particular, I would say that the 20.5 ka timing of your maximum ice extent is actually closer to observations than what is presented in these reconstructions (for instance, the maximum of the western half of the Laurentide ice sheet was achieved around that time Jackson et al. (2011); Lacelle et al. (2013)).

One thing that might be interesting to look at more is the causes of more major discrepancies in the model from geological observations. There are three main things that I would like to see comments on. I am guessing that these discrepancies are likely the result of biases in the climate model, but it would

be interesting to know more.

1) The Northwestern part of the Laurentide Ice Sheet, which covered Banks and Victoria islands, was one of the first places to deglaciate, but in your model it remains ice covered until after 8 ka.

2) An ice cap persists on the outer parts of the Grand Banks at the end of your simulation, a place that probably wasn't even glaciated during the MIS 2 glaciation. This seems like an odd place for an ice cap, considering it is below sea level and surrounded at all sides by the ocean.

3) Iceland remains ice covered through to the end of the simulation.

Best Regards,
Evan J. Gowan

**References**

Dalton, A.S., Margold, M., Stokes, C.R., Tarasov, L., Dyke, A.S., Adams, R.S., Allard, S., Arends, H.E., Atkinson, N., Attig, J.W., Barnett, P.J., Barnett, R.L., Batterson, M., Bernatchez, P., Borns, H.W., Breckenridge, A., Briner, J.P., Brouard, E., Campbell, J.E., Carlson, A.E., Clague, J.J., Curry, B.B., Daigneault, R.A., Dubé-Loubert, H., Easterbrook, D.J., Franzi, D.A., Friedrich, H.G., Funder, S., Gauthier, M.S., Gowan, A.S., Harris, K.L., Hétu, B., Hooyer, T.S., Jennings, C.E., Johnson, M.D., Kehew, A.E., Kelley, S.E., Kerr, D., King, E.L., Kjeldsen, K.K., Knaeble, A.R., Lajeunesse, P., Lakeman, T.R., Lamothe, M., Larson, P., Lavoie, M., Loope, H.M., Lowell, T.V., Lusardi, B.A., Manz, L., McMartin, I., Nixon, F.C., Occhietti, S., Parkhill, M.A., Piper, D.J., Pronk, A.G., Richard, P.J., Ridge, J.C., Ross, M., Roy, M., Seaman, A., Shaw, J., Stea, R.R., Teller, J.T., Thompson, W.B., Thorleifson, L.H., Utting, D.J., Veillette, J.J., Ward, B.C., Weddle, T.K., Wright, H.E., 2020. An updated radiocarbon-based ice margin chronology for the last deglaciation of the North American Ice Sheet Complex. Quaternary Science Reviews 234, 106223.

Gowan, E.J., Niu, L., Knorr, G., Lohmann, G., 2019. Geology datasets in North America, Greenland and surrounding areas for use with ice sheet models. Earth System Science Data 11, 375–391.

Hughes, A.L., Gyllencreutz, R., Lohne, Ø.S., Mangerud, J., Svendsen, J.I., 2016. The last Eurasian ice sheets–a chronological database and time-slice reconstruction, DATED-1. Boreas 45, 1–45.

Jackson, L.E., Andriashek, L.D., Phillips, F.M., 2011. Limits of successive middle and late Pleistocene continental ice sheets, interior plains of southern and central Alberta and adjacent areas, in: Ehlers, J., Gibbard, P.L., Hughes, P.D. (Eds.), Quaternary Glaciations - Extent and Chronology A Closer Look. Elsevier. volume 15 of *Developments in Quaternary Sciences*. chapter 45, pp. 575–589.

Lacelle, D., Lauriol, B., Zazula, G., Ghaleb, B., Utting, N., Clark, I.D., 2013. Timing of advance and basal condition of the Laurentide Ice Sheet during the last glacial maximum in the Richardson Mountains, NWT. Quaternary Research 80, 274–283.

Laske, G., Masters, G., 1997. A global digital map of sediment thickness. EOS Transactions 78, F483.

---

## Referee Comment (RC2)

**Review of: Climate and ice sheet evolutions from the last glacial maximum to the pre-industrial period with an ice sheet – climate coupled model by Quiquet et al**

Javier Blasco

*jablasco@ucm.es*

Quiquet and colleagues investigate the last deglaciation in the Northern Hemisphere using a coupled ice sheet - climate model. They use a climate model of intermediate complexity and a hybrid ice-sheet-shelf model. Overall, they simulate a deglaciation in good agreement with reconstructions. If they consider all the amplitude of the freshwater flux from the melted ice sheets, then the AMOC shuts down and is not able to recover. However, if they reduce these freshwater fluxes or consider additional mechanisms, such as brine rejection, then the AMOC can recover. Additional experiments show the sensitivity of their model to key parameters.

This is a very valuable effort and well suited for the scope of Climate of the Past. The manuscript is well written and easy to follow and I don't think that additional simulations are needed, but I have some comments and questions.

**General comments:**

**Reference experiment:**

I am curious about the selected parameters of the reference experiment. Were they chosen to simulate a realistic last glacial maximum (LGM) state? Have you tried to tune your present-day (PD) state? If so, what type of LGM state do you obtain/expect?

**Spin up**

You simulate separately the LGM state for the ice-sheet-shelf model and for the climate model. Then, your DGL experiment starts at 26 kyrBP, I guess to reach a sort of LGM equilibrium state for the coupled experiments. Do you obtain an equilibrated state? Have you tried to run an equilibrated LGM state with both models coupled from the start?

**Glacial isostatic adjustment:**

In P7 L204 it says:
*" We use a recent implementation of the last glacial maximum bathymetry at 21 kaBP (Lhardy et al., 2020), which is left unchanged for the duration of the experiments."*

When I first read this, I understood that the bathymetry was set constant for the whole experiment, including the deglaciation. However, in P14 L422 it is written:

*"At this time, the bedrock is still depressed below sea level over the northern most part of America but slowly returns to its present-day value."*

Indicating that the bedrock responds to changes in the load. I agree with the other reviewers opinion, that the GIA model needs to be described. Also, its potential implications in the retreat of part of the Eurasian and the Laurentide Ice Sheet should be discussed.

**Oceanic forcing**

You use in your ice-sheet model a linear melting law and you double the value for floating points in contact with the grounding line. I'm not very familiar with the most suited melting laws for the Greenland Ice Sheet, but I guess that in order to be more realistic, more complex processes should be taken into account, such as the plume formation or frontal ablation (*Slater et al., 2019, 2020*).

As I am more familiar with the Antarctic Ice Sheet, I know that a linear law is the least appropriate as it doesn't account for the positive feedback between the sub-shelf melting and the circulation in the ice-shelf cavity (*Favier et al., 2019*). Also, applying higher melting rates close to the grounding line for coarse resolution, as it is here, can overestimate the rates of grounding-line retreat (*Seroussi and Morlighem, 2018*). Perhaps, you may add one or two sentences on this point.

**Antarctic ice sheet**

P5L129: "*It is important to mention that only the Northern Hemisphere ice sheets are interactively simulated, while the Antarctic ice sheet topography and ice mask remains prescribed.*"
Prescribed to what? Present day? Last Glacial Maximum?

Also, if prescribed to LGM state, then you don't consider its potential sea-level rise which could accelerate grounding-line instabilities in your model.

**Brine rejection**

I found very interesting your results when you consider brine rejection in your model. I like this finding, maybe you can add a sentence on this in the abstract.

**Sensitivity experiments**

Do you run a new spin up for every sensitivity experiment? If so, how is it possible that all start at ~-100 msle in Figure 11?

**Technical comments:**

- You may cite here *Simms et al., 2019.*

- P8L227: *"With have performed ..."* Do you mean "We have performed..."?

- Figure 4: Color scale is missing in (a)

- Figure 5: If you draw temperature differences as in Figure 4 (b) then I would use the same color scale for consistency.

- Figure 11: Same as before. I would use the same colour for DGL_noFWF as in Figure 3 for consistency.

- P10 Table1: Although you explain in the manuscript what every parameter means, I would repeat it again in the description of the table.

**References:**

- Slater, D. A., Straneo, F., Felikson, D., Little, C. M., Goelzer, H., Fettweis, X., and Holte, J.: Estimating Greenland tidewater glacier retreat driven by submarine melting, The Cryosphere, 13, 2489–2509, https://doi.org/10.5194/tc-13-2489-2019, 2019.

- Slater, D. A., Felikson, D., Straneo, F., Goelzer, H., Little, C. M., Morlighem, M., Fettweis, X., and Nowicki, S.: Twenty-first century ocean forcing of the Greenland ice sheet for modelling of sea level contribution , The Cryosphere, 14, 985–1008, https://doi.org/10.5194/tc-14-985-2020, 2020.

- Favier, L., Jourdain, N. C., Jenkins, A., Merino, N., Durand, G., Gagliardini, O., Gillet-Chaulet, F., and Mathiot, P.: Assessment of sub-shelf melting parameterisations using the ocean–ice-sheet coupled model NEMO(v3.6)–Elmer/Ice(v8.3) , Geosci. Model Dev., 12, 2255–2283, https://doi.org/10.5194/gmd-12-2255-2019, 2019.

- Seroussi, H. and Morlighem, M.: Representation of basal melting at the grounding line in ice flow models, The Cryosphere, 12, 3085–3096, https://doi.org/10.5194/tc-12-3085-2018, 2018.

- Simms, A.R., Lisiecki, L., Gebbie, G., Whitehouse, P.L. and Clark, J.F., 2019. Balancing the last glacial maximum (LGM) sea-level budget. *Quaternary Science Reviews*, *205*, pp.143-153.

---

## Author Response (AR1)

**Response to referee 1, Dr. Evan J. Gowan**

1 Overview

Quiquet et al. present the results of a fully coupled ice sheet-climate model applied to the deglaciation of the Northern Hemisphere ice sheets from the Last Glacial Maximum. They use a climate model of intermediate complexity to couple in a bi-directional manner. Impressively, this is done at a very high frequency, with 5 years of ice sheet model for ever one year of simulated climate. The coupled model run produces a satisfactory representation of the deglaciation. One of the major findings is that the gradual input of freshwater in the ocean causes a shutdown of the AMOC. In some of their additional tests, they find that the AMOC does not shut down without this freshwater input. Other experiments showed only weak dependence on parameters related to surface and subglacial processes.

Overall, I enjoyed reading this paper, and found that it was well written. The experimental setup was well described (though note some of my comments below), and the implications for the ice sheet collapse on the overall climate was thorough. My comments below are mainly related to some parts of the modelling that should be elaborated a bit, and I do not think additional experiments are necessary. Therefore, I recommend relatively minor corrections.

Thank you for your review and your valuable comments.

2 Comments

2.1 Computational overhead

I am particularly impressed by 5/1 ratio of ice sheet to climate model years in this coupled setup. My question, though, is what is the computational costs of this model setup? If there is relatively low cost, then perhaps it could easily be adopted by other groups to investigate other problems, so this would be a good way to sell it.

In fact, we performed five experiments with synchronous, yearly, coupling (1/1 ratio): the reference experiments (DGL), three experiments with reduced freshwater flux (DGL_FWF/2, DGL_FWF/3 and DGL_noFWF) and finally one experiment with sinking brines parametrisation (DGL_brines). These experiments took, individually, about 31 days to complete.

The asynchronous experiments using a 5 to 1 ratio were roughly 5 times faster (since the ice sheet model is much less expensive than the rest of the climate model). They can be run in less than a week.

These numerical performance are probably not as good as the work performed with the LOVECLIM climate model (e.g. Heinemann et al., 2014 ; Choudhury et al., 2020) with which we share the main components (atmosphere, ocean and vegetation). This is due to the fact that the interactive downscaling used in our set-up to compute the surface mass balance decreases the performance of the standard model by about 40%.

We now provide this information in the text of the manuscript:
"The climate model computes about 850 years in 24 hours on a single core of an Intel® Xeon® CPU@3.70 GHz. The computational cost of the ice sheet model is negligible with respect to the rest of the climate model, while the interactive atmospheric downscaling decreases the performance by

about 40% compared to the standard climate model. The coupled synchronous experiments took roughly one month to complete while the asynchronous experiments were approximatively five times faster."

Of course, we encourage other groups interested by our tool to contact us to initiate a collaboration.

2.2 Ice sheet model resolution

Perhaps as part of the last point, I am curious as to why an ice sheet model resolution of 40 km is used. Is this a computational limitation? In experiments done by our group here at the Alfred Wegener Institute, we found that the trajectory of the ice sheet evolution can be radically different just by going from 40 km to 20 km resolution. In general, the computational expense of ice sheet models tends to be pretty low compared to climate models, so I think some discussion on this choice needs to be made.

It is true that the ice sheet model is relatively inexpensive compared to the rest of the climate system. In fact in our case, it is the atmospheric downscaling at the ice sheet model resolution that considerably increases the computational time (loss of about 40% in performance). Currently, using an higher ice sheet model resolution would results in a higher cost due to the downscaling. We could eventually imagine to downscale the atmospheric variables at a fixed spatial resolution (for example 40 km as here) and using simple spatial interpolation (e.g. bilinear) to transfer these variables to the ice sheet model at a higher resolution. However, we do not plan to implement this in the short term for mostly two reasons. First, given the non-linear nature of surface mass balance simple bilinear interpolation is not really adapted at the ice sheet margin. Second, we performed deglaciation GRISLI stand-alone experiments using two different spatial resolutions (40 km and 16 km) and the results were very similar. The evolutions of the ice sheet topography and velocity for selected snapshots, showing typical sheet deglacial geometry changes,  are shown in Fig. R1 and Fig. R2, respectively.

[Figure]

**Figure R1.** Simulated topography of the North American ice sheet for three selected snapshots (26, 12 and 8 kaBP) during the deglaciation. The top panels use a 40km grid resolution while the lower panels use a 16km grid resolution.

The forcing methodology is the same for both resolutions and follows the standalone forcing methodology of Quiquet et al. (2021). Here the IPSL-CM5A-LR model outputs from the PMIP3 database is used with a weighing factor for the millenial variability of 0.25 (see Quiquet et al., 2021). The colour scale is different for ice-free and ice-covered regions. The simulated ice sheet grounding line is represented by the red line while the black lines represent isocontours of ice sheet surface elevation (separated by 1000 metres).

[Figure]

**Figure R2.** Same as Fig. R1 but for the simulated vertically integrated velocity (m/yr) draped over the simulated topography.

**2.3 Glacial isostatic adjustment**

The GIA model used for the experiments is not described, so I would ask that this be added. Looking at Figure 9, the topography is depressed far more than in reality, which causes broad glacial lakes to form along the southern margin that are much bigger than during the actual deglaciation. This is one of the possible reasons (or even the main reason) that deglaciation was faster than expected.

We added a description of GIA:
"Glacial isostatic adjustment is accounted for in GRISLI using a elastic lithosphere - relaxed asthenosphere model (LeMeur and Huybrechts, 1996), with a relaxation time of the astherosphere of 3000 years."

The parameter of this simple GIA model are the same as in Quiquet et al. (2018). The fact that the bedrock topography is probably more depressed than in reality does not come from the GIA model but from the fact that the ice thickness is overestimated. The ice thickness overestimation can be due to an overestimation of the precipitation in the climate model and/or an underestimation of the ice sheet velocity (too low enhancement factor or a too high basal drag).

**2.4 Sediment thickness for basal sliding**

I don't understand why the Laske and Masters (1997) dataset was used to parameterize sediment distribution. That dataset is a map of Phanerozoic, undeformed sedimentary rock thickness for use in global seismology problems, and is vastly different to the distribution of unconsolidated

sediments that would be important for ice sheet sliding. There are places with sedimentary bedrock where there are no unconsolidated sediments (for instance in Ontario and central Manitoba), and there are places on the Precambrian Shield where there are thick unconsolidated sediments (for instance the Thelon Basin in Kivalliq). As a first order approximation, I guess you could assume that areas where the bedrock is Phanerozoic sedimentary rocks are more likely to be covered by unconsolidated sediment, so I don't think it would radically alter the results. However, I suggest in the future that a different dataset be used. Full disclosure, I have created such a dataset for North America (Gowan et al., 2019).

Thank you for pointing this out. We fully agree, sub-glacial sediments are probably key for ice dragging. However, it is unclear yet how they should be implemented in ice sheet model. Most of the time it is simply a local change of a friction parameter (as in here) but this question is also pretty much linked to the way basal dragging is implemented. Here for example we use a linear Coulomb friction law but it is possible that a strongly non-linear law should be used in some places (Gillet-Chaulet et al., 2016; Brondex et al., 2020). The degree of non-linearity of the friction law is probably related to the nature of the sediments below the ice sheet. However, we think that these are still open questions and we plan to explore them with our model in the future.

Related to this, thank you for pointing us to your paper. Historically, we use the map of Laske and Masters (1997) for its coverage of the whole globe but we agree that this could be updated with more recent / appropriate dataset. In Fig. R3 we show the sediment data as they are used in the model (a simple threshold value indicating an absence or a presence). We also show your dataset for the sediment distribution. Although your dataset displays a much higher spatial variability, they display an overall similar pattern. However, in some places key for deglaciation (e.g. present-day Hudson Bay) they have important differences.

Sub-glacial processes, including sediments, are certainly a very important direction for future research although it might not necessarily require a fully coupled climatic setup to be studied.

[Figure]

[Figure]

**Figure R3.** Left: sediment mask as it is used in the current version of the model, i.e. a 200 m threshold on Laske and Masters (1997). Brown areas indicate the presence of thick sediment leading to a smaller basal drag coefficient. Right: sediment distribution in Gowan et al. (2019) where the brown area shows the presence of thick sediment, yellow is discontinuous and white is an absence of sediment.

**2.5 Spinup time**

A 200 kyr spinup is used to initialize the ice sheet to the LGM state. I'm wondering why such a long spinup was necessary, considering that during the last interglacial (about 100 kyr before the LGM), there were essentially no ice sheets in the Northern Hemisphere except for perhaps part of the Greenland Ice Sheet. Even the Eurasian Ice Sheets were probably almost non-existent just 15 kyr before the LGM (Hughes et al., 2016). Would such a long spinup affect the results?

Our spinup methodology cannot be compared with a real glacial inceptions. In particular, for the whole duration of the spinup, the climate forcing remains constant (glacial conditions simulated by *i*LOVECLIM using prescribed ice sheet). In this climate forcing, there is a very low precipitation rate, in particular over the domes of the ice sheets. Ice sheet build-up with a more realistic transient climate evolution might be faster.

An other aspect is that we wanted to start our experiments with equilibrated ice sheets to avoid drifts in our ice sheet model. In doing so, it is easier to quantify the impact of climate change. Of course this is an approximation of the reality since the ice sheets were probably never "equilibrated" with the climate.

Some variables in the model requires long integrations to reach an equilibrium under constant climate forcing. The internal temperature needs a few tens of thousand of years, starting from a linear vertical profile in Greenland for example. An other variable that needs long integrations is the hydraulic head since we only rely on a simple advection/diffusion scheme, and this variable is coupled with the velocity field: a larger hydraulic head is associated with a larger velocity and the heat due to friction produces melt water.

We acknowledge the fact that the assumption that the ice sheets are in equilibrium with the glacial climate has consequences. However we do not see how this could be properly quantified except by running a full glacial-interglacial cycle, which remains currently a numerical challenge for us.

We added the following in the discussion section:
"Lastly, we run deglaciation experiments starting from 26 kaBP assuming that the Northern Hemisphere ice sheets were in equilibrium with the simulated glacial climate. However, the last glacial maximum ice sheets were the results of the long previous glacial period starting from the last glacial inception. Ideally, it would have been best to perform a transient coupled experiment covering this period of time in order to have a more realistic ice sheet states. Notably, slow evolving ice sheet variables such as glacial isostasy or internal temperatures are expected to be affected by a transient spin-up instead of a constant glacial spin-up. However, this remains currently a numerical challenge to perform such a transient spin-up."

**2.6 Comparison with geological data**

There is a section that compares the modeled results with some ice sheet reconstructions. I think this is fine, but don't feel too bad that you don't match things exactly, since the margin chronology in North America is in the process of being revised (Dalton et al., 2020). In some places the timing of advance and retreat is being revised by thousands of years. In particular, I would say that the 20.5 ka timing of your maximum ice extent is actually closer to observations than what is presented in these reconstructions (for instance, the maximum of the western half of the Laurentide ice sheet was achieved around that time Jackson et al. (2011); Lacelle et al. (2013)).

One thing that might be interesting to look at more is the causes of more major discrepancies in the model from geological observations. There are three main things that I would like to see comments on. I am guessing that these discrepancies are likely the result of biases in the climate model, but it would be interesting to know more.

1) The Northwestern part of the Laurentide Ice Sheet, which covered Banks and Victoria islands, was one of the first places to deglaciate, but in your model it remains ice covered until after 8 ka.

2) An ice cap persists on the outer parts of the Grand Banks at the end of your simulation, a place that probably wasn't even glaciated during the MIS 2 glaciation. This seems like an odd place for an ice cap, considering it is below sea level and surrounded at all sides by the ocean.

3) Iceland remains ice covered through to the end of the simulation.

Your point 1 & 2 can be largely explained by the biases in the climate model.

We present in Fig. R4a a map of the absolute annual near-surface temperature for a reference pre-industrial experiment (with fixed ice sheets). This pre-industrial simulation is run in a similar way than the deglaciation experiment. For example it uses a LGM oceanic bathymetry and a closed Bering Strait as this are fixed climate model features for the deglaciation experiments. The Northern Hemisphere topography and ice mask are nonetheless here at their present-day reference value for GRISLI, i.e. ETOPO1 (Amante and Eakins, 2009) when ice free and Bamber et al. (2013) when ice-covered. Fig. R4b is the absolute annual near-surface temperature for ERA5 climatological mean over 1979-2008. Fig. R4c is the temperature difference (b-a).

We also show the annual mean total precipitation simulated by the model (Fig. R4d), in the CRU-CL-v2 dataset (New et al., 2002, Fig. R4e) and the ration between the two (d/e, Fig. R4f).

The regions you mention in your point 1 & 2 show a cold bias associated with an overestimation of the precipitation.

In addition, there is an other factor that can explain your point 2 & 3. The ice caps over the Grand Banks and Iceland represent an important topographic anomaly with respect to the standard pre-industrial climate. This topographic anomaly drastically increases the annual precipitation rate (greater than 3 m/yr). Also, there is a strong albedo feedback than leads to little melt in this areas even at the end of our simulations at 0 kaBP. We acknowledge that is counter intuitive to retain an ice cap over the Grand Banks. However, the simulated ice thickness there is large enough (>1000 m) to maintain a grounded ice sheet over the ocean that is relatively shallow (no deeper than 200 metres).

We have added the following in the manuscript:
"The chronology and pattern of the deglaciation is largely affected by the biases in the climate model. We present these bi- ases in term of mean annual temperature and total precipitation rate in Fig. 10. To construct this figure we use a reference pre-industrial experiment (with fixed ice sheets), performed with a similar setup to the deglaciation experiments. Notably, this pre-industrial experiment uses the same last glacial oceanic bathymetry with a closed Bering Strait. The Northern Hemisphere topography and ice mask are nonetheless at their present-day reference value for GRISLI (Amante and Eakins, 2009; Bamber et al., 2013). The model presents a cold bias associated with an overestimation of the precipitation in the northwestern part of the North American continent. This explains why this region of the North American ice sheet deglaciates much later

than its eastern sector where a warm bias is present. Also, Grand Banks and Iceland remain ice covered at the end of the simulation where the model is generally too cold and too wet. More generally, the climate model tends to overestimate the precipitation over mountainous areas which can induce a positive feedback over some ice caps such as Iceland, Grand Banks, the Ellesmere Islands and the Scandinavian mountains."

[Figure]

**Figure R4.** (a) Simulated annual near-surface air temperature for a pre-industrial climate experiment using the model configuration used for the deglaciation experiment (i.e. with a LGM ocean bathymetry) but with a present-day topography and ice mask for the Northern Hemisphere. (b) Annual near-surface air temperature for the ERA5 climatological mean over 1979-2008. (c) Temperature difference a-b. (d) Simulated annual total precipitation rate for the same pre-industrial experiment. (e) CRU-CL-v2 annual total precipitation rate. (f) Precipitation ratio d/e.

**References:**

Amante, C. and Eakins, B.: ETOPO1 1 Arc-Minute Global Relief Model: Procedures, Data Sources and Analysis, NOAA Technical Memorandum NESDIS NGDC-24, National Geophysical Data Center, NOAA, 2009.

Bamber, J. L., Griggs, J. A., Hurkmans, R. T. W. L., Dowdeswell, J. A., Gogineni, S. P., Howat, I., Mouginot, J., Paden, J., Palmer, S., Rignot, E., and Steinhage, D.: A new bed elevation dataset for Greenland, The Cryosphere, 7, 499–510, https://doi.org/10.5194/tc-7-499-2013, 2013.

Brondex, J., Gillet-Chaulet, F., and Gagliardini, O.: Sensitivity of centennial mass loss projections of the Amundsen basin to the friction law, The Cryosphere, 13, 177–195, https://doi.org/10.5194/tc-13-177-2019, 2019.

Choudhury, D., Timmermann, A., Schloesser, F., Heinemann, M., and Pollard, D.: Simulating Marine Isotope Stage 7 with a coupled climate–ice sheet model, Clim. Past, 16, 2183–2201, https://doi.org/10.5194/cp-16-2183-2020, 2020.

Gillet-Chaulet, F., Durand, G., Gagliardini, O., Mosbeux, C., Mouginot, J., Rémy, F., and Ritz, C.: Assimilation of surface velocities acquired between 1996 and 2010 to constrain the form of the basal friction law under Pine Island Glacier, Geophys. Res. Lett., 43, 10311–10321, doi10.1002/2016GL069937, 2016.

Heinemann, M., Timmermann, A., Elison Timm, O., Saito, F., and Abe-Ouchi, A.: Deglacial ice sheet meltdown: orbital pacemaking and $CO_2$ effects, Clim. Past, 10, 1567–1579, https://doi.org/10.5194/cp-10-1567-2014, 2014.

Le Meur, E. and Huybrechts, P.: A comparison of different ways of dealing with isostasy: examples from modeling the Antarctic ice sheet during the last glacial cycle, Annals of Glaciology, 23, 309–317, https://doi.org/10.3189/S0260305500013586, 1996.

New, M., Lister, D., Hulme, M., and Makin, I.: A high-resolution data set of surface climate over global land areas, Climate Res., 21, 1–25, https://doi.org/10.3354/cr021001, 2002.

Quiquet, A., Dumas, C., Ritz, C., Peyaud, V., and Roche, D. M.: The GRISLI ice sheet model (version 2.0): calibration and validation for multi-millennial changes of the Antarctic ice sheet, Geosci. Model Dev., 11, 5003–5025, https://doi.org/10.5194/gmd-11-5003-2018, 2018.

Quiquet, A., Dumas, C., Paillard, D., Ramstein, G., Ritz, C., and Roche, D. M.: Deglacial Ice Sheet Instabilities Induced by Proglacial Lakes, Geophysical Research Letters, 48, e2020GL092 141, https://doi.org/10.1029/2020GL092141, 2021.

**Response to referee 2, Dr. Javier Blasco**

Quiquet and colleagues investigate the last deglaciation in the Northern Hemisphere using a coupled ice sheet - climate model. They use a climate model of intermediate complexity and a hybrid ice-sheet-shelf model. Overall, they simulate a deglaciation in good agreement with reconstructions. If they consider all the amplitude of the freshwater flux from the melted ice sheets, then the AMOC shuts down and is not able to recover. However, if they reduce these freshwater fluxes or consider additional mechanisms, such as brine rejection, then the AMOC can recover. Additional experiments show the sensitivity of their model to key parameters.

This is a very valuable effort and well suited for the scope of Climate of the Past.
The manuscript is well written and easy to follow and I don't think that additional simulations are needed, but I have some comments and questions.

Thank you for your time revising our manuscript. We answer your comments in the following and we changed the manuscript accordingly.

General comments:

Reference experiment:

I am curious about the selected parameters of the reference experiment. Were they chosen to simulate a realistic last glacial maximum (LGM) state? Have you tried to tune your present-day (PD) state? If so, what type of LGM state do you obtain/expect?

In fact, this work is a result of a few years of development and calibration, for both the climate and ice sheet models. From the climate model side, over the years, we made a few modifications from the Goosse et al. (2010) original core of the climate model (ice mask, surface energy budget, parameters related to the downscaling of precipitation, etc.) with the aim of reducing known important model biases under pre-industrial conditions (notably: warm bias in North America, cold bias in the Arctic, overestimation of precipitation over mountainous regions). If some of the biases were sometimes reduced we were nonetheless not able to suppress them all. A map of the temperature and precipitation biases is shown in this response (Fig. R4) and now included in the manuscript. We did not specifically try to tune the LGM climate state, we simply checked that the LGM vs. PI temperature change was in a relatively good agreement with published literature (e.g. Kageyama et al., 2020).

However, for the coupling parameters and coupling strategy (melt coefficient in the surface and sub-shelf melt models, the sub-grid albedo, ageing of the snow albedo, etc.), we tuned both the PI and the LGM in parallel. These model choices were first tested under a PI climate and assessed with the simulated Greenland ice sheet volume. However, the Greenland ice sheet offers only a relatively weak constraint for these parameters given its extension with respect to the atmospheric model grid size. Melt models that produced a closer agreement with the present-day ice sheet volume were producing largely too small Northern Hemisphere ice sheets at the LGM.

Finally we did not consider the ice sheet model parameters as tuning parameters for the coupled model. These were calibrated independently with offline ice sheet model simulations of four glacial-interglacial cycles of the Antarctic ice sheet, as in Quiquet et al. (2018). Even if some ice sheet parameters could influence the coupled response (e.g. a more dynamic ice sheet could in

principle have a larger extent) the climate model biases seem in our case much more influential on the simulated ice sheet state at the first order.

Spin up

You simulate separately the LGM state for the ice-sheet-shelf model and for the climate model. Then, your DGL experiment starts at 26 kyrBP, I guess to reach a sort of LGM equilibrium state for the coupled experiments. Do you obtain an equilibrated state? Have you tried to run an equilibrated LGM state with both models coupled from the start?

With our methodology we do not claim to reach an equilibrium at 21 kaBP since we use transient climate forcing from 26 to 21 kaBP. However, we expect to reach some consistency between the simulated climate and the simulated topography. We agree with what you suggest: the consistency would have been even better with an equilibrium simulation coupled from the start. In fact, we perform such an experiment more recently using a coupling frequency of 1:10 yr and a duration of 1 kyr for the climate model (10 kyr for the ice sheet model). The simulated ice sheets at the end of the equilibrium is shown in Fig. R5. They do show some differences but they are generally similar.

The reason why we did not use this approach from the start is that it is still much more computationally expensive than the one we followed. In our approach, we use only one long (3000 yr) climate equilibrium under glacial conditions to perform plenty of short (100 yr) experiments with various formulations of the melt model. The different climatological surface mass balance are then used to force offline the ice sheet model (inexpensive).

[Figure]

**Figure R5. (a)** Simulated ice sheet topography used as initial condition for the start of the coupled experiments, i.e. after the 200 kyr offline ice sheet spin-up. **(b)** Simulated ice sheet topography simulated after a 1000 yr coupled experiment under perpetual glacial conditions starting from (a), using an acceleration factor of 10 (10 kyr are simulated by the ice sheet model). The colour scale is different for ice-free and ice-covered regions. The simulated ice sheet grounding line is represented by the red line while the black lines represent isocontours of ice sheet surface elevation (separated by 1000 metres).

We added few elements in the discussion section:
"Lastly, we run deglaciation experiments starting from 26 kaBP assuming that the Northern Hemisphere ice sheets were in equilibrium with the simulated glacial climate. However, the last glacial maximum ice sheets were the results of the long previous glacial period starting from the last glacial inception. Ideally, it would have been best to perform a transient coupled experiment covering this period of time in order to have a more realistic ice sheet states. Notably, slow evolving ice sheet variables such as glacial isostasy or internal temperatures are expected to be affected by a transient spin-up instead of a constant glacial spin-up. However, this remains currently a numerical challenge to perform such a transient spin-up."

Glacial isostatic adjustment:

In P7 L204 it says: " We use a recent implementation of the last glacial maximum bathymetry at 21 kaBP (Lhardy et al., 2020), which is left unchanged for the duration of the experiments."

When I first read this, I understood that the bathymetry was set constant for the whole experiment, including the deglaciation. However, in P14 L422 it is written: "At this time, the bedrock is still depressed below sea level over the northern most part of America but slowly returns to its present-day value."

Indicating that the bedrock responds to changes in the load. I agree with the other reviewers opinion, that the GIA model needs to be described. Also, its potential implications in the retreat of part of the Eurasian and the Laurentide Ice Sheet should be discussed.

GIA is accounted for in the ice sheet model with a simple Elastic Lithosphere – Relaxed Asthenophere (ELRA) model (LeMeur and Huybrechts, 1996). The ice sheet model is also forced by transient eustatic sea level rise (Waelbroeck et al., 2002).

This has been clarified in the text, in the model description section:
"Glacial isostatic adjustment is accounted for in GRISLI using a elastic lithosphere - relaxed asthenosphere model (LeMeur and Huybrechts, 1996), with a relaxation time of the astherosphere of 3000 years."
And later, in the experimental setup section:
"On the ice sheet model side, in addition to the climate forcings, an other forcing is the transient eustatic sea level reconstruction from Waelbroeck et al. (2002)."

Glacial isostasy largely explains the grounding line instability that occurs at the southern margin of the North American ice sheet. This is discussed in Quiquet et al. (2021), now referenced in the manuscript. Glacial isostasy can also play a role for the Eurasian ice sheet since the bedrock in the Barents-Kara region is more depressed than today with retrograde slopes from the grounding line. This favours the marine ice sheet instability that occurs at 14.5 kaBP in our experiments. We added this in the manuscript:
"Such instability is favoured by the depressed bedrock, with a ~300 m deepening in the Kara sea with respect to the present-day bathymetry, resulting in steeper retrograde slopes."

However, the bathymetry of the oceanic model is left unchanged in the course of the deglaciation. We have clarified this, first in the method section:
"For the experiments presented here, changes in the ice sheet size do not affect the global ocean volume. The bathymetry in the oceanic model remains thus constant."

And later for the description of the experimental setup:
"For the oceanic model, we use a recent implementation of the last glacial maximum bathymetry at 21 kaBP (Lhardy et al., 2021), which is left unchanged for the duration of the experiments."

Oceanic forcing

You use in your ice-sheet model a linear melting law and you double the value for floating points in contact with the grounding line. I'm not very familiar with the most suited melting laws for the Greenland Ice Sheet, but I guess that in order to be more realistic, more complex processes should be taken into account, such as the plume formation or frontal ablation (Slater et al., 2019, 2020).

As I am more familiar with the Antarctic Ice Sheet, I know that a linear law is the least appropriate as it doesn't account for the positive feedback between the sub-shelf melting and the circulation in the ice-shelf cavity (Favier et al., 2019). Also, applying higher melting rates close to the grounding line for coarse resolution, as it is here, can overestimate the rates of grounding-line retreat (Seroussi and Morlighem, 2018). Perhaps, you may add one or two sentences on this point.

We fully agree with this comment although the Greenland ice sheet glacier frontal melt is certainly a too fine scale process to be correctly represented in a 40 km grid resolution using oceanic fields at a 3° resolution.

We have added the following in the manuscript when we discuss the mass loss partitioning:
"The lesser importance of the sub-shelf melt rate for the first phase of the deglaciation could arise from the simple model we use to represent this process. Notably, we use a linear melting rate dependency on temperature change, while a quadratic dependency could best reproduce this process (Favier et al., 2019). A quadratic dependency would result in more sensitive melt rate changes to temperature changes."

We plan to implement an alternative sub-shelf melt model at the interface between GRISLI and iLOVECLIM. However, the main driver for ice sheet retreat in our experiment is surface mass balance, at least until 12.8 kaBP. After this date, sub-shelf melt rate becomes important only becauzse grounding line instabilities have been triggered. These instabilities are not triggered by the artificially high grounding line melting rate since the experiment with higher sub-shelf melt displays a similar ice sheet evolution.

We added the following in the discussion:
"Second, we have used a very simple parametrisation for sub-shelf melt when alternative parametrisations display a better agreement with complex sub-shelf cavity oceanic models (Favier et al., 2019). This process is key for the future of Antarctic ice sheet (Seroussi et al., 2020) and could be equally important for the deglaciation of marine based sectors of the Northern Hemisphere ice sheets (Petrini et al., 2018; Clark et al., 2020). For this reason, we plan to implement an alternative sub-shelf melt model at the interface between GRISLI and iLOVECLIM. However, in our experiments, the main driver for ice sheet retreat is surface mass balance, at least until 12.8 kaBP. After this date, sub-shelf melt rate becomes important only because grounding line instabilities have been triggered. These instabilities do not seem to be triggered by an artificially high grounding line melting rate since the experiment with higher sub-shelf melt displays a very similar ice sheet evolution. This results could be revisited with a more complex sub-shelf model."

Antarctic ice sheet

P5L129: "It is important to mention that only the Northern Hemisphere ice sheets are interactively simulated, while the Antarctic ice sheet topography and ice mask remains prescribed."
Prescribed to what? Present day? Last Glacial Maximum?

At the Last Glacial Maximum, following the PMIP protocol. This is now explicitly mentioned.

Also, if prescribed to LGM state, then you don't consider its potential sea-level rise which could accelerate grounding-line instabilities in your model.

The ice sheet model is forced with transient eustatic sea level reconstruction from Waelbroeck et al. (2002), which include the contribution from Antarctica, which is probably about 10 m (e.g. Whitehouse et al., 2012; Briggs et al., 2014). For earlier version of the coupled model, we performed sensitivity experiments with other eustatic sea level reconstructions (Lambeck et al., 2014) with no significant differences. The reason is that for our experiments the ice sheet evolution is primarily driven by climate forcing, not sea level forcing.

Brine rejection

I found very interesting your results when you consider brine rejection in your model. I like this finding, maybe you can add a sentence on this in the abstract.

We have done so:
"The inclusion of a parametrisation for the sinking of brines around Antarctica also produces an abrupt recovery of the Atlantic meridional overturning circulation, absent in the reference experiment."

Sensitivity experiments

Do you run a new spin up for every sensitivity experiment? If so, how is it possible that all start at ~-100 msle in Figure 11?

This is an important point indeed, and no, we did not. This is a questionable modelling choice but we wanted to quantify the different climate trajectories starting from a common initial condition. We have added this precision in the manuscript, in the experimental setup section:
"All the experiments, including the sensitivity experiments with perturbed parameter values, use the same spun-up climate and ice sheet states."

It is true that an other alternative would have been to run new spun-up ice sheets for the different sensitivity experiments. This would have resulted on sometimes important differences for the ice sheet geometry at the start of the coupled experiment. One reason why we did not choose this approach is that it would have lead to a poorer agreement with the geologically reconstructions during the glacial period.

Technical comments:

- You may cite here Simms et al., 2019.

We have added the reference.

- P8L227: "With have performed ..." Do you mean "We have performed..."?

Yes, thanks for noticing.

- Figure 4: Color scale is missing in (a)

I do not understand: (a) and (b) use the same colour scale. Both sub-panel show absolute annual near-surface air temperature, for the LGM (a) and for PI (b).

- Figure 5: If you draw temperature differences as in Figure 4 (b) then I would use the same color scale for consistency.

Fig. 4b is the simulated temperature for the pre-industrial, not a temperature difference. It might be best to not use the same colour scale for the absolute temperature field and a temperature difference. But maybe you meant Fig. 4d, which is the LGM temperature anomaly with respect to PI? Fig. 5 is the temperature difference from the two simulated PI, with and without the ice sheet melt freshwater feedback on the ocean. The range of Fig. 5, with positive and negative values, is largely different from the range of Fig. 4d (mostly negative).

- Figure 11: Same as before. I would use the same colour for DGL_noFWF as in Figure 3 for consistency.

We have done so.

- P10 Table1: Although you explain in the manuscript what every parameter means, I would repeat it again in the description of the table.

Information added.

References:

- Slater, D. A., Straneo, F., Felikson, D., Little, C. M., Goelzer, H., Fettweis, X., and Holte, J.: Estimating Greenland tidewater glacier retreat driven by submarine melting, The Cryosphere, 13, 2489–2509, https://doi.org/10.5194/tc-13-2489-2019, 2019.
- Slater, D. A., Felikson, D., Straneo, F., Goelzer, H., Little, C. M., Morlighem, M., Fettweis, X., and Nowicki, S.: Twenty-first century ocean forcing of the Greenland ice sheet for modelling of sea level contribution , The Cryosphere, 14, 985–1008, https://doi.org/10.5194/tc-14-985-2020, 2020.
- Favier, L., Jourdain, N. C., Jenkins, A., Merino, N., Durand, G., Gagliardini, O., Gillet-Chaulet, F., and Mathiot, P.: Assessment of sub-shelf melting parameterisations using the ocean–ice-sheet coupled model NEMO(v3.6)–Elmer/Ice(v8.3) , Geosci. Model Dev., 12, 2255–2283, https://doi.org/10.5194/gmd-12-2255-2019, 2019.
- Seroussi, H. and Morlighem, M.: Representation of basal melting at the grounding line in ice flow models, The Cryosphere, 12, 3085–3096, https://doi.org/10.5194/tc-12-3085-2018, 2018.

- Simms, A.R., Lisiecki, L., Gebbie, G., Whitehouse, P.L. and Clark, J.F., 2019. Balancing the last glacial maximum (LGM) sea-level budget. Quaternary Science Reviews, 205, pp.143-153.

**References:**

Briggs, R. D., Pollard, D., and Tarasov, L.: A data- constrained large ensemble analysis of Antarctic evolu- tion since the Eemian, Quaternary Sci. Rev., 103, 91–115, https://doi.org/10.1016/j.quascirev.2014.09.003, 2014.

Kageyama, M., Harrison, S. P., Kapsch, M.-L., Lofverstrom, M., Lora, J. M., Mikolajewicz, U., Sherriff-Tadano, S., Vadsaria, T., Abe-Ouchi, A., Bouttes, N., Chandan, D., Gregoire, L. J., Ivanovic, R. F., Izumi, K., LeGrande, A. N., Lhardy, F., Lohmann, G., Morozova, P. A., Ohgaito, R., Paul, A., Peltier, W. R., Poulsen, C. J., Quiquet, A., Roche, D. M., Shi, X., Tierney, J. E., Valdes, P. J., Volodin, E., and Zhu, J.: The PMIP4 Last Glacial Maximum experiments: preliminary results and comparison with the PMIP3 simulations, Clim. Past, 17, 1065–1089, https://doi.org/10.5194/cp-17-1065-2021, 2021.

Lambeck, K., Rouby, H., Purcell, A., Sun, Y., and Sambridge, M.: Sea level and global ice volumes from the Last Glacial Maximum to the Holocene, Proceedings of the National Academy of Sciences, 111, 15 296, https://doi.org/10.1073/pnas.1411762111, 2014.

Le Meur, E. and Huybrechts, P.: A comparison of different ways of dealing with isostasy: examples from modeling the Antarctic ice sheet during the last glacial cycle, Annals of Glaciology, 23, 309–317, https://doi.org/10.3189/S0260305500013586, 1996.

Quiquet, A., Dumas, C., Paillard, D., Ramstein, G., Ritz, C., and Roche, D. M.: Deglacial Ice Sheet Instabilities Induced by Proglacial Lakes, Geophysical Research Letters, 48, e2020GL092 141, https://doi.org/10.1029/2020GL092141, 2021.

Waelbroeck, C., Labeyrie, L., Michel, E., Duplessy, J. C., McManus, J. F., Lambeck, K., Balbon, E., and Labracherie, M.: Sea-level and deep water temperature changes derived from benthic foraminifera isotopic records, Quaternary Science Reviews, 21, 295–305, https://doi.org/16/S0277-3791(01)00101-9, 2002.

Whitehouse, P. L., Bentley, M. J., and Le Brocq, A. M.: A deglacial model for Antarctica: geological constraints and glacio- logical modelling as a basis for a new model of Antarctic glacial isostatic adjustment, Quaternary Sci. Rev., 32, 1–24, https://doi.org/10.1016/j.quascirev.2011.11.016, 2012.